# A cross-species assessment of behavioral flexibility in compulsive disorders

Nabil Benzina [1✉], Karim N'Diaye[1], Antoine Pelissolo[2,3], Luc Mallet[1,2,4] & Eric Burguière [1✉]

Lack of behavioral flexibility has been proposed as one underlying cause of compulsions, defined as repetitive behaviors performed through rigid rituals. However, experimental evidence has proven inconsistent across human and animal models of compulsive-like behavior. In the present study, applying a similarly-designed reversal learning task in two different species, which share a common symptom of compulsivity (human OCD patients and *Sapap3* KO mice), we found no consistent link between compulsive behaviors and lack of behavioral flexibility. However, we showed that a distinct subgroup of compulsive individuals of both species exhibit a behavioral flexibility deficit in reversal learning. This deficit was not due to perseverative, rigid behaviors as commonly hypothesized, but rather due to an increase in response lability. These cross-species results highlight the necessity to consider the heterogeneity of cognitive deficits in compulsive disorders and call for reconsidering the role of behavioral flexibility in the aetiology of compulsive behaviors.

[1] Institut du Cerveau, ICM, Inserm U 1127, CNRS UMR 7225, Sorbonne Université, 47 bd de l'Hôpital, 75013 Paris, France. [2] Assistance Publique-Hôpitaux de Paris, DMU IMPACT, Département Médical-Universitaire de Psychiatrie et d'Addictologie, Hôpitaux Universitaires Henri Mondor-Albert Chenevier, Université Paris-Est Créteil, 40 rue de Mesly, 94000 Créteil, France. [3] INSERM U955, IMRB, 8 rue du Général Sarrail, 94010 Créteil cedex, France. [4] Department of Mental Health and Psychiatry, Global Health Institute, University of Geneva, 9 Chemin des Mines, 1202 Geneva, Switzerland. ✉email: nabil.benzina@unige.ch; eric.burguiere@icm-institute.org

Behavioral flexibility is a cognitive function which refers to the ability to dynamically adjust behavior to a changing environment[1]. A lack of behavioral flexibility results in rigid behaviors, echoing pathological conditions observed in OCD patients who are resistant to change. Thus, behavioral flexibility impairments have been proposed has one of the causes of compulsive behaviors[2]. However, there is no consensus whether such a deficit exists in OCD as inconsistencies were found between studies using reversal learning[3,4], task switching[5,6], or intra/extra-dimensional set shifting[7,8] paradigms. Beyond methodological considerations such as small sample sizes[9,10], the clinical heterogeneity of OCD patients may have contributed to these discrepant results[11–15]. In contrast, dysfunctional activations of OCD patients' prefrontal regions, in particular the orbitofrontal cortex (OFC), during performance of flexibility tasks has been more consistently reported[16,17]. Similar neurobiological observations have been recently made in the *Sapap3* knock-out mutant mice (*Sapap3* KO), the current predominant genetic model of compulsive-like behavior[18]. These genetically engineered mice lack the SAP90/PSD95-associated protein 3, a postsynaptic scaffolding protein mainly expressed in the striatum[19]. This mutation results in the expression of excessive grooming behaviors, which can be defined as compulsive-like given the associated neurophysiological impairments of the prefronto-striatal circuits[18,20], including the OFC[21], and the reduction of grooming behavior after chronic administration of fluoxetine, a first-line treatment for OCD patients[18]. Regarding behavioral flexibility, two recent studies[22,23] have challenged this model in a spatial reversal learning task and found that *Sapap3* KO mice had impaired performances compared to controls after reversal event, although the type of deficit differed with increased perseveration found in one study[23] but not the other[22]. Interestingly, both studies found that these deficits were not correlated with the severity of compulsive-like grooming, suggesting that compulsivity and flexibility dimensions may be distinctly affected in this model. Moreover, one of the studies identified a lack of flexibility only in a subgroup of *Sapap3* KO mice[23]. This result highlights inherent model heterogeneity, which echoes the clinical heterogeneity observed in OCD patients.

To improve the comparison of human and animal results, similar experimental procedures across species are indispensable in order to ensure comparable task parameters and psychometric properties[24,25], and, hence, comparable results[26]. In case of reversal learning tasks, it has been demonstrated that the neurobiological processes may differ according to sensory modalities, impeding translational value in approaches using different stimulus modality across species[27]. Hence, proper data transposition from animal models to humans and vice versa in tasks assessing behavioral flexibility is hindered[28,29] by the fact that flexibility assessment in animal models mostly rely on spatial discrimination tasks[22,23,30–33] while in humans visual discrimination tasks are most commonly applied.

Therefore, in order to study the involvement of behavioral flexibility in compulsive behaviors in both OCD patients and the *Sapap3* KO mice, we have developed an innovative, high throughput behavioral setup for mice that allows us to reliably test individual subjects in a non-spatial visual reversal learning task through multiple reversal blocks, as it is commonly performed in human studies. We furthermore ensured the correct interpretation of our results by recruiting large samples of well characterized and selected subjects in both species, thereby enabling us to investigate intra-group variability in our analyses.

## Results

**Compulsiveness is unrelated to behavioral flexibility as assessed by the reversal learning paradigm.** In both species, we applied a similarly-designed reversal learning task to assess their behavioral flexibility (Fig. 1 and Supplementary Discussion for

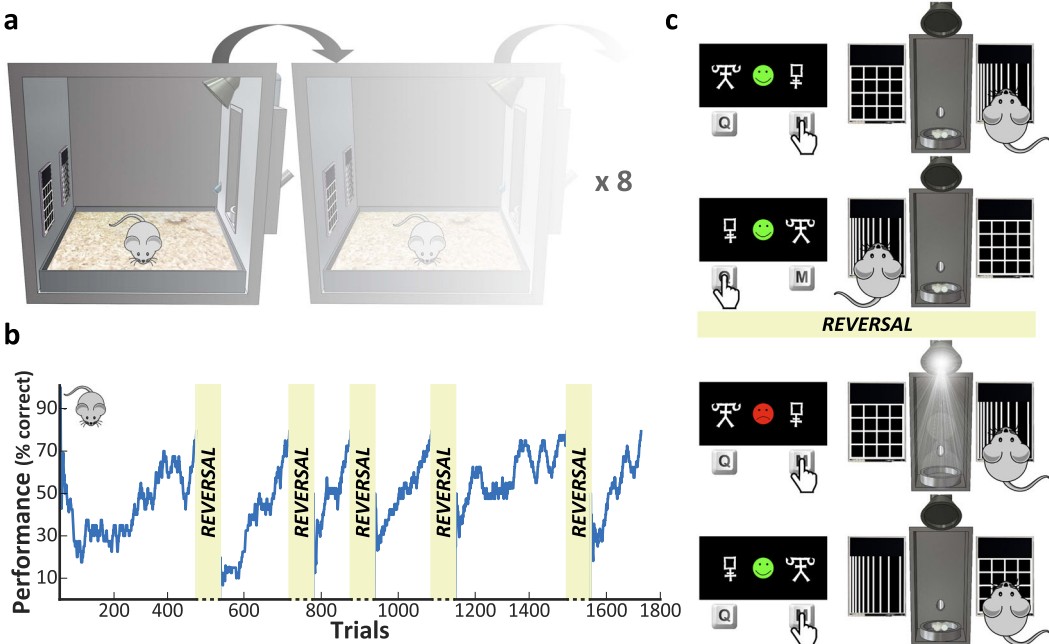

**Fig. 1 Experimental design and apparatus of the reversal learning task. a** Illustration of the behavioral apparatus. Up to 8 operant conditioning chambers run in parallel with mice living and working with minimal human intervention. Each operant chambers was equipped with capacitive touch-sensitive screens (left), pellet and water dispenser (right), and LED lights (top). **b** Example of one WT mouse performance (smoothed over a 40-trial sliding window) across five reversals. **c** Design of the human (left) and the mouse (right) versions of the reversal learning task. On each trial, the subject had to make a choice between two different stimuli displayed on the screens. Depending on their choice, positive (for correct response) or negative (for incorrect response) feedback was provided. When the subject had learned the correct association, the reward contingencies were reversed without notice (see "Methods" section for details).

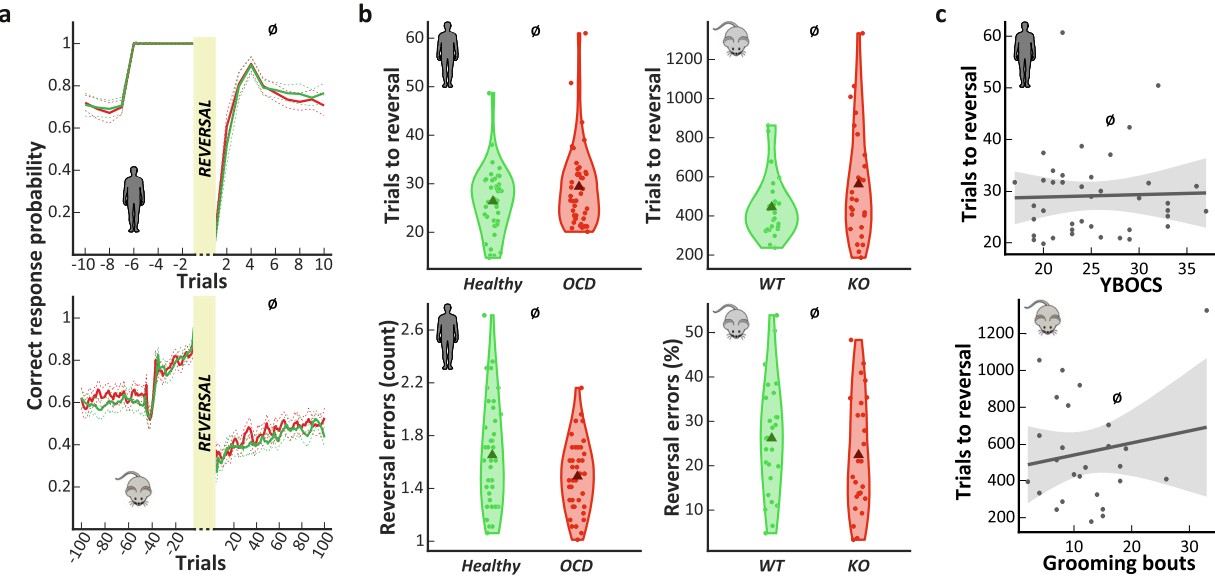

**Fig. 2 Compulsivity is not related to behavioral flexibility. a** Changes in correct response probability around a reversal. Top: for humans with 10 trials around the reversal. Red line: OCD patients. Green line: healthy subjects. Bottom: for mice with 100 trials around the reversal. Red line: Sapap3 KO mice. Green line: WT mice. The data were smoothed using Savitzky–Golay algorithm for both species. **b** No difference was found between groups for both humans (left, $n = 40$ per group) and mice (right, $n = 26$ per group), neither when considering the number of trials needed to reach the reversal criterion (top) or the number of reversal errors (bottom). Triangle: group mean. Dot: individual mean. Ø: $BF_{10} < 1$. **c** Top: In OCD patients, the disease severity assessed by the Y-BOCS does not predict the number of trials needed to reach the reversal criterion. Dark line: linear fit. Gray area: confidence interval. Bottom: In Sapap3 KO mice, compulsive grooming severity assessed by the number of grooming bouts initiated (over a 10-min period) does not predict the number of trials needed to reach the reversal criterion. Dark line: linear fit. Gray area: confidence interval.

the underlying rationale). We observed that the performance profile after a reversal event was similar between compulsive subjects and their controls in the two species ($BF_{Inclusion} < 1$ for group factor, Fig. 2a and Supplementary Tables S1 and S2). The number of trials needed to reach reversal criterion (Fig. 2b, top) did not differ between compulsive and control groups, neither for human subjects ($BF_{10} = 0.64$, $d = 0.27$ [0.04 0.77]), nor mice ($BF_{10} = 0.7$, $d = 0.33$ [−0.1 0.92]). Similarly, no significant group differences were found in the number of reversal errors (Fig. 2b, bottom), neither in humans ($BF_{10} = 1.5$, $d = 0.35$ [0.07 0.88]), nor mice ($BF_{10} = 0.44$, $d = 0.25$ [−0.2 0.83]).

The comparison of other behavioral parameters, such as spontaneous strategy changes (SSC) probability and SSC errors (see definition in "Methods" section), support this lack of difference between groups for both species (Table 1).

In OCD patients, correlation analysis showed that disease severity and task performance were not related (Fig. 2c, top and Table 2). Likewise, we found no correlation in mice between grooming level and the main behavioral parameters (Fig. 2c, bottom and Table 2).

**Distinct subgroups of OCD patients and *Sapap3* KO mice exhibit a behavioral flexibility deficit.** In OCD patients ($n = 40$), depression/anxiety levels and antidepressants did not influence task performance (Table 2 and see Supplementary Notes for more results relative to medication). When we assessed the effect of symptom subtypes (such as "checking", "washing", "hoarding", etc...) on task performance, only severity of the "checking" subtype was positively correlated to an increased number of trials needed to reach the reversal criterion (Fig. 3a and Table 2). We thus conducted another analysis by separating a subgroup of twenty-one OCD "checkers" with predominantly checking symptoms from the other OCD patients. The three resulting distinct groups ($n = 21$ OCD "checkers", $n = 19$ OCD "non-checkers" and $n = 40$ healthy subjects) were not different in terms

of demographic characteristics (Table 3). In terms of clinical characteristics, the group of OCD "checkers" showed a higher rate of comorbid anxiety disorder compared to the two other groups (Table 3).

As for OCD patients, we attempted to determine a similar heterogeneity in the *Sapap3* KO mice ($n = 26$). Indeed, some innovative studies have found evidence for inter-individual variability of cognitive traits in animal models[34,35], even in *Sapap3* KO mice[23], pointing out the necessity to consider this variability in animal studies. Thus, we performed a two-step cluster analysis which identified two clusters within the *Sapap3* KO mice (silhouette measure = 0.6, $BIC_{2\ clusters} = 92.82$; compared to $BIC_{1\ cluster} = 96.13$ and $BIC_{3\ clusters} = 109.66$). The resulting ΔBIC of −3.31 indicated a positive evidence in favor of this clustering. The same procedure was applied to WT controls ($n = 26$) without the detection of separate clusters. Among the different variables used to perform the analysis favouring the two-cluster solution, the SSC probability was identified as the most important variable, followed by the reversal error proportion, the number of trials needed to reach reversal criterion and the SSC perseverative errors (importance values = 1, 0.79, 0.42, and 0.19, respectively). Three distinct groups resulted from this clustering procedures: the WT mice ($n = 26$); the "unimpaired" *Sapap3* KO mice ($n = 14$), defined as overlapping with WT; and the "impaired" *Sapap3* KO mice ($n = 12$), defined as distant from the WT (Fig. 3c). We confirmed our clustering analysis by performing a stepwise discriminant analysis: overall, 71.2% of mice were correctly labeled with 83.3% of the "impaired" KO mice correctly classified (16.7% were classified as WT) and 50% for the "unimpaired" KO mice (50% classified as WT). These results were consistent with those of the two-step cluster analysis: the "unimpaired" *Sapap3* KO mice cluster was closer to the WT mice cluster (Fig. 3c) with a moderate agreement between the two analyses (κ = 0.53, 95% CI: [0.32 0.73], $p < .0005$). The two *Sapap3* KO subgroups, which resulted from the two-step cluster

**Table 1 Humans and mice behavioral parameters in the reversal learning task.**

**Humans**

| | Healthy (H) | OCD | $BF_{10}$ | Checkers (C) | Non-checkers (NC) | $BF_{10}^{ANOVA}$ |
|---|---|---|---|---|---|---|
| Trials in acquisition phase | 19.79 (7.29) | 24.09 (9.89) | 2.29 | 28.13 (11.59) | 19.62 (4.74) | 47.88: C > NC = H |
| Trials to reversal | 26.09 (6.84) | 29.09 (8.43) | 0.64 | 31.87 (10.05) | 26.02 (4.8) | 3.98: C > NC = H |
| Reversal errors | 1.64 (0.4) | 1.48 (0.27) | 1.5 | 1.46 (0.26) | 1.51 (0.28) | 0.72 |
| SSC probability as % | 4.1 (3.17) | 5.2 (4.68) | 0.41 | 6.47 (5.44) | 3.8 (3.27) | 1.19: C > NC = H |
| SSC errors | 0.17 (0.19) | 0.17 (0.16) | 0.17 | 0.21 (0.17) | 0.12 (0.15) | 0.35 |
| SCAPE probability as % | 54.79 (22.53) | 61.82 (15.26) | 0.6 | 61.86 (14.78) | 61.77 (16.17) | 0.33 |
| SCAPE errors | 1.28 (0.16) | 1.29 (0.21) | 0.19 | 1.31 (0.25) | 1.26 (0.15) | 0.16 |

**C57BL/6J mice**

| | WT (W) | *Sapap3* KO | $BF_{10}$ | Impaired KO (I) | Unimpaired KO (U) | $BF_{10}^{ANOVA}$ |
|---|---|---|---|---|---|---|
| Trials in acquisition phase | 464.66 (182.23) | 438.39 (248.19) | 0.22 | 560.08 (261.76) | 334.07 (187.31) | 2.35: U < I = W |
| Trials to reversal | 437.7 (156.89) | 554.39 (290.37) | 0.7 | 746.87 (278.55) | 389.4 (181.61) | >100: W = U < I |
| Reversal errors as % | 25.84 (12.8) | 22.08 (12.97) | 0.44 | 11.1 (5.17) | 31.49 (9.71) | >100: I < U = W |
| SSC probability as % | 44.19 (3.65) | 44.4 (4.09) | 0.21 | 48.1 (1.88) | 41.22 (2.36) | >100: U < W < I |
| SSC errors | 0.57 (0.11) | 0.45 (0.09) | 27.74 | 0.41 (0.08) | 0.48 (0.1) | >100: I < U < W |

Mean values (with standard deviation) of main behavioral parameters of the reversal learning task in both species. A $BF_{10}$ greater than one is in favor of a difference and vice versa. The further the $BF_{10}$ is from one, the greater the evidence. A $BF_{10}$ greater than 3 (or lower than $1/3$) is commonly considered a significant evidence. For JZS ANOVA, the results of post hoc tests are given after the $BF_{10}$ value with "=" indicating an absence of difference. *SSC* spontaneous strategy change, *SCAPE* strategy change after a probabilistic error.

---

**Table 2 Correlation scores between clinical and task parameters.**

**OCD patients**

| | Trials to reversal | Reversal errors |
|---|---|---|
| Trials in acquisition phase | $BF_{10}$ = 0.28, r = 0.14 [−0.18 0.42] | |
| Fluoxetine-equivalent dose | $BF_{10}$ = 0.24, r = −0.105 [−0.39 0.21] | $BF_{10}$ = 0.31, r = 0.16 [−0.16 0.43] |
| YBOCS | $BF_{10}$ = 0.2, r = 0.03 [−0.28 0.33] | $BF_{10}$ = 0.2, r = 0.02 [−0.29 0.32] |
| BDI | $BF_{10}$ = 0.34, r = −0.17 [−0.44 0.15] | $BF_{10}$ = 0.21, r = 0.05 [−0.25 0.35] |
| STAI-A | $BF_{10}$ = 0.22, r = 0.08 [−0.23 0.37] | $BF_{10}$ = 0.21, r = 0.03 [−0.25 0.35] |
| STAI-B | $BF_{10}$ = 0.22, r = −0.08 [−0.37 0.23] | $BF_{10}$ = 0.2, r = 0 [−0.3 0.3] |
| OCI-R total score | $BF_{10}$ = 0.2, r = −0.02 [−0.31 0.29] | $BF_{10}$ = 0.33, r = 0.17 [−0.15 0.44] |
| OCI-R checking subscore | $BF_{10}$ = 2.96, r = 0.37 [0.06 0.6] | $BF_{10}$ = 0.2, r = 0 [−0.3 0.3] |
| OCI-R washing subscore | $BF_{10}$ = 0.47, r = −0.22 [−0.48 0.1] | $BF_{10}$ = 0.23, r = 0.1 [−0.21 0.38] |
| OCI-R order subscore | $BF_{10}$ = 0.27, r = −0.13 [−0.41 0.18] | $BF_{10}$ = 0.2, r = 0.04 [−0.27 0.33] |
| OCI-R hoarding subscore | $BF_{10}$ = 0.2, r = 0.04 [−0.27 0.33] | $BF_{10}$ = 0.24, r = -0.11 [−0.39 0.2] |
| OCI-R obsession subscore | $BF_{10}$ = 0.2, r = 0.02 [−0.29 0.32] | $BF_{10}$ = 0.53, r = 0.23 [−0.09 0.49] |
| OCI-R neutralization subscore | $BF_{10}$ = 0.25, r = −0.12 [−0.4 0.2] | $BF_{10}$ = 0.42, r = 0.2 [−0.11 0.47] |

| | | OCI-R checking |
|---|---|---|
| SSC probability as % | | $BF_{10}$ = 4.54, r = 0.4 [0.1 0.62] |
| SSC errors | | $BF_{10}$ = 0.58, r = 0.24 [−0.08 0.5] |
| SCAPE probability as % | | $BF_{10}$ = 0.38, r = -0.19 [−0.46 0.13] |
| SCAPE errors | | $BF_{10}$ = 0.42, r = 0.2 [−0.11 0.47] |

**Sapap3 KO mice**

| | Trials to reversal | Reversal errors as % |
|---|---|---|
| Number of grooming bouts | $BF_{10}$ = 0.32, r = 0.16 [−0.23 0.49] | $BF_{10}$ = 0.65, r = −0.29 [−0.59 0.11] |
| Time spent grooming as % | $BF_{10}$ = 0.29, r = −0.12 [−0.46 0.27] | $BF_{10}$ = 0.49, r = −0.25 [−0.55 0.15] |

In patients, only the checking dimension influences the performance in our task; whereas in *Sapap3* KO mice, there is no influence of grooming severity. A $BF_{10}$ greater than one is in favor of a correlation and vice versa. The further the $BF_{10}$ is from one, the greater the evidence. A $BF_{10}$ greater than 3 (or lower than $1/3$) is commonly considered a significant evidence. *OCI-R* obsessive compulsive inventory revised, *YBOCS* Yale-Brown obsessive-compulsive scale, *BDI* Beck depression inventory, *STAI* Spielberger's state (A)–trait (B) anxiety inventory, *BIS-10* Barratt impulsivity scale, *fNART* French national adult reading test, *SSC* spontaneous strategy change, *SCAPE* strategy change after a probabilistic error.

---

analysis and which were confirmed via a stepwise discriminant analysis, were similar in terms of weight and grooming level (Table 3), showed comparable locomotor activity and task engagement (Supplementary Fig. S1), and had no identified genealogical difference (Supplementary Fig. S2). Noteworthy, we conducted the same clustering procedure on the human data with comparable results than the ones observed in mice. Two clusters were found in OCD patients with an impaired subgroup of 7 patients, 6 of them being checkers; and only one cluster for healthy controls (see Supplementary Notes for details). These results validated the relevance of using the clinical dimension of "checking" symptoms as a subgroup splitting factor.

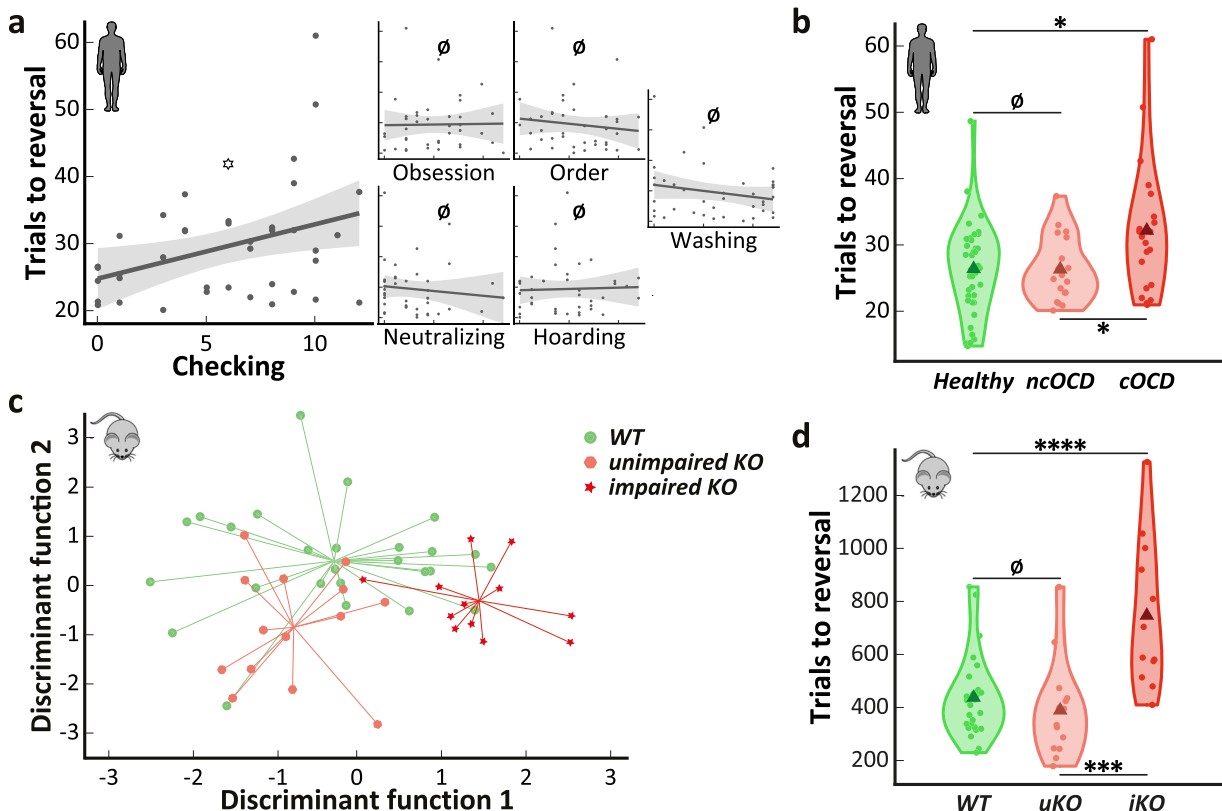

**Fig. 3 Only a subgroup of OCD patients and *Sapap3* KO mice needed a higher number of trials to reach the reversal criterion. a** Only the severity of the checking symptoms (measured by the OCI-R checking subscore) predicts the number of trials needed to reach the reversal criterion. The higher the checking symptoms severity, the higher the number of trials. $n = 40$ OCD patients. Dark line: linear fit. Gray area: confidence interval. **b** 21 OCD patients with predominant checking symptoms ("checkers" subgroup, red) were segregated from the others (19 "non-checkers" and 40 healthy controls subgroups, pink and green, respectively). Only OCD "checkers" patients were impaired in terms of number of trials needed to reach the reversal criterion. ncOCD: "non-checkers". cOCD: "checkers". Triangle: group mean. Dot: individual mean. **c** A two-step cluster analysis using four behavioral parameters (number of trials to reversal, reversal errors, SSC probability and SSC errors) found two distinct clusters within the Sapap3 KO mice and was confirmed by a stepwise discriminant analysis. The intersection point of the lines indicates the group's centroid. **d** Only the "impaired" KO mice ($n = 12$, red) needed more trials to reach the reversal criterion compared to the other "unimpaired" KO ($n = 14$, pink) and WT mice ($n = 26$, green). uKO: "unimpaired" KO mice. iKO: "impaired" KO mice. Triangle: group mean. Dot: individual mean. Ø: $BF_{10} < 1$. $^*BF_{10} < 3$. $^*BF_{10} \geq 3$. $^{**}BF_{10} \geq 10$. $^{***}BF_{10} \geq 30$. $^{****}BF_{10} \geq 100$.

To assess behavioral flexibility in these different subgroups of both species, we systematically compared their performance in terms of the number of trials to reach reversal criterion. We detected a difference in the number of trials to reach criterion between OCD "checkers", OCD "non-checkers" and healthy controls ($BF_{10} = 3.98$, $\eta^2 = 0.11$, Fig. 3b, Table 1). Indeed, a post-hoc analysis revealed that OCD "checkers" needed more trials than both OCD "non-checkers" ($BF_{+0} = 4.66$, $d = 0.74$ [0.08 1.25]) and healthy controls ($BF_{+0} = 9.32$, $d = 0.67$ [0.14 1.17]). We detected no difference between the latter two groups ($BF_{10} = 0.28$, $d = 0.01$ [−0.5 0.5]). Comorbid anxiety disorder or gender effect were absent in humans (Supplementary Tables S3 and S4). Similarly, we found a difference in the number of trials to reach reversal criterion between "impaired" *Sapap3* KO, "unimpaired" *Sapap3* KO mice and WT controls ($BF_{10} > 100$, $\eta^2 = 0.35$, Fig. 3d). Indeed, in an according post-hoc analysis, "impaired" KO mice needed more trials than WTs ($BF_{10} > 100$, $d = 1.37$ [0.54 2.17]), and we detected no difference between "unimpaired" KO mice and WTs ($BF_{10} = 0.43$, $d = 0.29$ [−0.35 0.82]).

**Reversal learning deficit is explained by higher response lability rather than perseveration.** Considering that only checking symptoms were associated with a reversal learning impairment in OCD patients, we investigated which behavioral trait could

explain this deficit. We first observed that the number of reversal errors did not correlate with severity of checking symptoms, suggesting that OCD "checkers" did not express greater perseverative behaviors than healthy subjects (Table 2). This was confirmed by an absence of a group effect on the number of reversal errors when comparing the OCD "checkers", OCD "non-checkers" and healthy controls subgroups ($BF_{10} = 0.72$, $\eta^2 = 0.06$, Fig. 4a, left). In mice, a group effect relative to the proportion of reversal errors ($BF_{10} > 100$, $\eta^2 = 0.34$, Fig. 4a right) was shown but we found that "impaired" *Sapap3* KO mice performed fewer reversal errors than WTs ($BF_{10} = 53.84$, $d = 1.51$ [0.35 1.93], Fig. 4a, right). These results suggested that behavioral flexibility deficits observed in subgroups of compulsive subjects were not explained by greater perseveration. On the contrary, we observed a positive correlation between the severity of "checking" symptoms and the probability of spontaneous strategy change (SSC); i.e., changing its response despite a positive feedback (Table 2). These results suggested that OCD "checkers" had a high response lability as identified through an elevated SSC probability. The subgroup analysis in both species supported this result, with differences of SSC probability observed between the three subgroups either in humans ($BF_{10} = 1.19$, $\eta^2 = 0.08$, Fig. 4b left) or mice ($BF_{10} > 100$, $\eta^2 = 0.41$, Fig. 4b right). OCD "checkers" had greater SSC probability than both healthy controls ($BF_{+0} = 3.51$, $d = 0.53$ [0.08 1.02]) and OCD "non-checkers" ($BF_{+0} = 2.22$,

**Table 3 Demographic and clinical characteristics of the samples.**

**Humans**

| | Healthy (H) | BF₁₀ | OCD | BF₁₀ | Checkers (C) | Non-checkers (NC) | BF₁₀ |
|---|---|---|---|---|---|---|---|
| Sample size | 40 | | 40 | | 21 | 19 | |
| Gender (M/F) | 15/25 | | 15/25 | | 10/11 | 5/14 | |
| Hand (R/L) | 35/5 | | 34/6 | | 16/5 | 18/1 | |
| Age in years | 40.28 (13.59) | 0.26 | 40.15 (13.22) | 0.44 | 41.9 (12.78) | 38.21 (13.77) | |
| Education in years | 5.43 (2.19) | 0.2 | 5.15 (2.69) | 0.34 | 5.1 (3.24) | 5.21 (1.99) | |
| fNART IQ | 108.18 (6.76) | 0.18 | 107.4 (6.87) | 0.15* | 106.81 (7.12) | 108.05 (6.71) | |
| BDI | 0.8 (1.34) | >100 | 12.53 (6.56) | 0.13* | 11.05 (5.09) | 14.16 (7.68) | >100*: H < C = NC |
| BIS-10 | 45 (9.89) | 0.23 | 47.03 (11.75) | 0.14* | 48.57 (12.14) | 45.32 (11.38) | |
| STAI-A | 40.40 (4.38) | >100 | 62.60 (13.07) | 0.21* | 62.81 (10.08) | 62.37 (16.04) | >100*: H < C = NC |
| STAI-B | 36.05 (5.97) | >100 | 65.38 (9.05) | | 64.76 (5.65) | 66.05 (11.88) | >100*: H < C = NC |
| OCI-R | | | | | | | |
| Total score | 3.88 (4.23) | >100 | 30.43 (8.86) | | 31.19 (7.85) | 29.58 (10) | >100*: H < C = NC |
| Order subscore | 1.45 (1.88) | >100 | 6.28 (3.06) | | 6.14 (3.14) | 6.42 (3.04) | >100*: H < C = NC |
| Washing subscore | 0.53 (1.06) | >100 | 6.45 (4.59) | | 4.43 (4.23) | 8.68 (3.96) | >100*: H < C < NC |
| Checking subscore | 0.35 (0.89) | >100 | 5.68 (3.87) | | 8.52 (2.23) | 2.53 (2.65) | >100*: H < NC < C |
| Hoarding subscore | 1.1 (1.37) | >100 | 3.85 (2.93) | | 4.29 (2.26) | 3.37 (3.53) | >100*: H < C = NC |
| Obsession subscore | 0.38 (0.77) | >100 | 5.53 (2.97) | | 5.1 (2.41) | 6 (3.5) | >100*: H < C = NC |
| Neutralization subscore | 0.18 (0.5) | >100 | 2.65 (2.86) | | 2.71 (3.02) | 2.58 (2.76) | >100*: H < C = NC |
| YBOCS | | | 25.25 (5.29) | | 24 (4.84) | 26.63 (5.54) | 0.85 |
| Taking an antidepressant | | | 28 | | 16 | 12 | 0.74 |
| with an AAP | | | 4 | | 2 | 2 | 0.24 |
| with a benzodiazepine | | | 7 | | 4 | 3 | 0.3 |
| Fluoxetine-equivalent in mg/d | | | 53.73 (57.65) | | 46.74 (47.88) | 61.45 (67.34) | 0.4 |
| Psychiatric comorbidities | | | | | | | |
| Anxiety disorder | | | 11 | | 9 | 2 | 6.27 |
| Eating disorder | | | 1 | | 0 | 1 | |
| Age at onset | | | 15.23 (5.88) | | 15.05 (5.64) | 15.42 (6.28) | 0.31 |
| Duration in years | | | 24.92 (13.95) | | 26.86 (15.45) | 22.79 (12.14) | 0.43 |

**C57BL/6J mice**

| | WT (W) | BF₁₀ | Sapap3 KO | BF₁₀ | Impaired KO (I) | Unimpaired KO (U) | BF₁₀ |
|---|---|---|---|---|---|---|---|
| Sample size | 26 | | 26 | | 12 | 14 | |
| Age in days | 200.12 (12.23) | 0.25 | 199.35 (12.29) | | 199.67 (14.9) | 199.07 (10.13) | 0.16* |
| Weight in g | 35.47 (4.99) | >100 | 29.71 (2.88) | | 29.13 (3.32) | 30.21 (2.46) | >100*: W > I = U |
| Number of grooming bouts | 5.38 (3.75) | >100 | 12.12 (7.05) | | 14.58 (8.55) | 10 (4.82) | >100*: W < I = U |
| Time spent grooming in % | 11.76 (19.18) | 3.11 | 25 (20.43) | | 29.01 (24.2) | 21.56 (16.72) | 1.71*: W < I = U |

Mean values (with standard deviation) of demographic and clinical characteristics for human and mice populations. A BF₁₀ greater than one is in favor of a difference (and vice versa). The further the BF₁₀ is from one, the greater the evidence (and vice versa). A BF₁₀ greater than 3 (or lower than 1/3) is commonly considered a significant evidence. The * indicates a significant evidence. The * indicates a JZS ANOVA BF₁₀ with the results of post hoc tests given after the BF₁₀ value; "=" indicating an absence of difference. AAP atypical antipsychotic, OCI-R obsessive compulsive inventory revised, YBOCS Yale-Brown obsessive-compulsive scale, BDI Beck depression inventory, BIS-10 Barratt impulsivity scale, BIS-10 anxiety inventory, STAI Spielberger's State (A)-trait (B) anxiety inventory, fNART French national adult reading test.

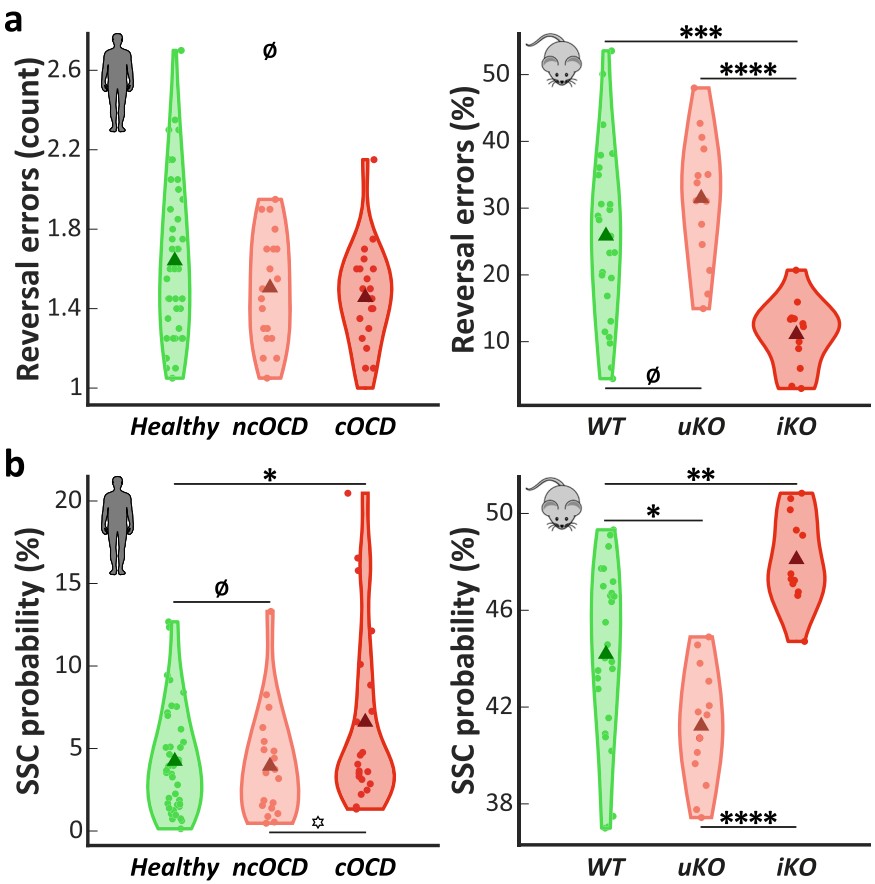

**Fig. 4 An excessive response lability underlies the reversal learning deficit. a** Deficits of behavioral flexibility found in both OCD "checkers" patients ($n = 21$) and "impaired" KO mice ($n = 12$) subgroups were not explained by a greater perseveration (in term of reversal errors). **b** Instead, both OCD "checkers" and "impaired" KO mice showed an increased SSC probability compared to other groups ($n = 19$ "non-checkers", 40 healthy controls, 14 "unimpaired" KO and 26 WT mice), suggesting a higher response lability. ncOCD: "non-checkers". cOCD: "checkers". uKO: "unimpaired" KO mice. iKO: "impaired" KO mice. Triangle: group mean. Dot: individual mean. Ø: $BF_{10} < 1$. $^{*}BF_{10} < 3$. $^{*}BF_{10} \geq 3$. $^{**}BF_{10} \geq 10$. $^{***}BF_{10} \geq 30$. $^{****}BF_{10} \geq 100$.

$d = 0.59$ [0.06 1.1]). The healthy controls and OCD "non-checker" groups did not differ from each other ($BF_{10} = 0.29$, $d = 0.09$ [−0.41 0.57]) and no comorbid anxiety disorder or gender effect was found (Supplementary Tables S5 and S6). The same results were obtained in mice with "impaired" KO mice showing greater SSC probability than WTs ($BF_{10} = 24.64$, $d = 1.22$ [0.33 1.82]), and "unimpaired" KO mice showing less SSC probability than WTs ($BF_{10} = 5.24$, $d = 0.91$ [0.12 1.45]). In the same line, WTs perseverated more after SSC, with more SSC errors than "impaired" KO mice ($BF_{10} > 100$, $d = 1.61$ [0.62 2.24]) (Table 1). Importantly, these difference in response lability (expressed here as an SSC probability) were observed only in a reversal context and not during the acquisition phase, both in OCD "checkers" and "impaired" KO mice (see Supplementary Notes).

## Discussion

This cross-species study assessed the role of behavioral flexibility in compulsive behaviors through a reversal learning task conducted in both humans and mice. We showed that in both species compulsive subjects do not form a homogeneous group. Taken as a whole, neither the human nor rodent compulsive groups showed differences in task performance compared to their controls. Thus, the severity of compulsive behavior per se was not a predictor of performance in our reversal learning task. In contrast, when heterogeneity within groups was taken into account, we identified in both species a subgroup with strong behavioral flexibility deficit in our task. Importantly, this deficit was

independent of compulsive behavior severity but rather linked to checking symptoms in patients. In addition, we found in both species that, contrary to what we would expect, the deficit of behavioral flexibility observed in some subgroups was not underpinned by excessive perseverative behavior after reversal but rather by greater response lability. Taken together, our results from a cross-species perspective do not support a link between compulsion and behavioral flexibility. Instead, they suggest that another dimension, excessive response lability, found in subgroups of compulsive subjects has an effect on behavioral flexibility.

Our study also emphasizes the importance of considering clinical subtypes within OCD patients, as encouraged by other recent studies[36,37]. The fact that we found deficit only in OCD checking patients is in line with a recent meta-analysis demonstrating that the neuropsychological profile of checking patients is more disrupted than in other OCD patients, with major impairment in planning/problem solving, response inhibition and set-shifting (and therefore in executive functioning overall)[38]. Similarly, the identification of only a subgroup of *Sapap3* KO mice impaired in our reversal learning task echoes what was reported in a recent study by Manning and colleagues[23] which also found that only a subgroup of these mice was impaired in a spatial reversal learning task although no difference in grooming level was highlighted.

The excessive response lability observed in some subgroups for both species, particularly in humans, echoes the results of computational studies performed by Kanen and colleagues[39] that

reported reduced stimulus stickiness in OCD patients; and the results found by Hauser and colleagues[40] that showed a lower win-stay probability in OCD patients compared to control subjects with a lower perseveration parameter in their model. However, unlike our study, the different OCD subtypes were not taken into account and therefore the influence of the "checking" dimension in their result cannot be excluded. This excessive response lability could also be seen as a specific form of perseveration, the subject having difficulties in suppressing the previous association long after the reversal. However, OCD patients have also decision making[41] and information sampling[42] impairments specific to situations of uncertainty. In that respect, this increased response lability could be induced by an increased level of uncertainty provoked by the reversal event. This assumption makes sense when considering the isolated subgroup of "checker" patients displaying a higher degree of uncertainty. In mice, we cannot conclude that the isolated subgroup of *Sapap3* KO mice is analogous to the "checking" subtype of OCD. However, considering that it is not uncommon for a patient to present a hybrid compulsive symptomatology mixing both compulsive checking and washing[43], one can imagine that the impaired subgroup of mice also presents a mix of compulsive grooming and checking behaviors. Obviously, in mice these checking behaviors are not directly observable, but it would be interesting to test if uncertainty monitoring and checking behaviors are also affected in *Sapap3* KO mice. Indeed, one could expect that abnormal increase of uncertainty after reversal would provoke the excessive lability of their behavior (e.g., with mice over-checking if the previously rewarded stimulus is still valid). Another dimension which could affect both compulsivity and flexibility is the overexpression of habitual behaviors. Some recent studies have favored this idea in OCD patients[44] and *Sapap3* KO mice[45] but others dampened[46] or rejected this hypothesis[47]. Thus, more evidence will be needed to fully understand the implication of habits in compulsive behaviors and flexibility. Finally, it cannot be ruled out that the deficit found is linked to an impairment that is not task-specific, such as an attentional impairment, whether primary[38] or secondary to obsessive activity in patients (particularly for checking patients, as the reversal increases uncertainty and thus possibly the obsessive doubt)[48]. It would thus be important in the future to simultaneously assess attention, like other cognitive dimensions, to better characterize the underlying mechanisms of this observed behavioral inflexibility.

A potential limitation of our study that can be pointed out is the difference in terms of medication between the two species. Indeed, while our *Sapap3* KO mice were free of any pharmacological treatment that could alter their performance, OCD patients were largely on serotoninergic treatment. It is thus logical to think that the results obtained may reflect the effect of the treatment and not their disorder. However, we were able to show the absence of influence of serotoninergic medication on performance in our task with no difference between medicated and unmedicated patients. Further, it has been shown that chronic administration of a selective serotonin reuptake inhibitor reduces perseveration and promotes a win-stay strategy in this type of task[49], which is contrary to our results. We can therefore be confident of the cross-species validity of these findings.

Our cross-species results favor the heterogeneity of cognitive deficits observed in compulsive disorders and stress the importance of also considering this heterogeneity in animal models. Indeed, even if inbred mouse lines share identical genetic background, this is not necessarily stable over time and may result in the emergence of new phenotypic traits due to a genetic drift. However, it has been shown that the C57BL/6J strain is one of the strains least susceptible to this effect[50]. Furthermore, we could not identify any genealogical specificity for the impaired KO mice

subgroup. Another hypothesis for the heterogeneity we observed in our animal model could be of epigenetic and/or environmental origin. It has been shown for example that phenotypic variability can emerge from variations in epigenetic regulation[35,51–53]. Examples include studies on genetically homogeneous WT C57BL/6J mice showing inter-individual variability in the expression of flexible behavior underpinned by variability in serotonin levels within the OFC[54]. As we could not identify any subgroup in our WT mice based on the task performance, such inter-individual variability could only be a risk factor whose sole interaction with the *Sapap3* KO mutation leads to an impairment.

The use of Bayesian statistics is another strength of our study which allowed us to formally support the absence of prespecified differences, notably in terms of perseveration but also to quantify the weight of evidence in favor or against those between-group differences. This methodological strength also points out a limitation to the interpretation of our results, especially those obtained with humans. Indeed, the differences highlighted between the checking patients and the healthy subjects are all supported by a Bayes Factor lower than 10 reflecting a low to substantial evidence but far from being decisive with a moderate effect size. This speaks toward the need to replicate these results on a larger sample than the one included in this study.

In conclusion, we found that compulsive behavior is not necessarily associated with a deficit in behavioral flexibility. In contrast, this study proposes that a behavioral flexibility deficit, only observed in a subset of compulsive subjects, may result from excessive response lability rather than perseveration, both in humans and mice.

## Methods
**Participants and animal subjects**. OCD patients were recruited through an online advertisement posted on a patient association's website (AFTOC) and among a cohort of severe patients followed in the psychiatric department of Albert Chenevier Hospital. Healthy comparison subjects were recruited through an online advertisement posted on an information web site dedicated to cognitive research (RISC). Diagnoses and co-morbidity were established by an experienced clinician with the French version of the Mini International Neuropsychiatric Interview (MINI v5[55]). Exclusion criteria were defined as follows: actual major depressive episode, bipolar disorder, acute or chronic psychosis, substance abuse or dependency including alcohol, epilepsy, cerebral injury, or other neurological problems. To assess severity and clinical subtypes of obsessive-compulsive (OC) symptoms, the Yale-Brown Obsessive-Compulsive Scale (YBOCS[56]) was administered only for OCD patients, and the Obsessive Compulsive Inventory Revised (OCI-R[57]) was used to measure all participants' OC characteristics. Forty patients were included in this study. They were diagnosed with OCD according to the DSM-V criteria and had a score greater than or equal to 16 on the YBOCS. Among those, fifteen patients displayed contamination/washing symptoms, thirteen aggressive/checking symptoms, eight predominant aggressive/checking symptoms associated with contamination/washing symptoms and four had predominantly obsessive thoughts, mainly religious/mental rituals. The mean age at onset of OCD symptoms was 15.23 (±5.881) years old and the mean illness duration was 24.92 (±13.951) years. Depression, trait/state anxiety and impulsivity were assessed, respectively, with the short version of the Beck Depression Inventory (BDI[58]), the Spielberger's State-Trait Anxiety Inventory (STAI[59]) and the Barratt impulsivity scale (BIS-10[60]) in their French version. Among the patients taking part in this study, twenty-eight were free of any psychiatric comorbidity, eleven had a comorbid anxiety disorder (essentially general and social anxiety disorder) and one had an eating disorder. Considering psychotropic medication, twenty-eight patients took an antidepressant drug alone or combined with antipsychotics or mood stabilizer and the remaining patients were medication-free. The current pharmacological treatment was converted to dose-equivalent fluoxetine for each patient[61]. Forty healthy control subjects, free of any current psychiatric or neurological disorder and subsequent medications, were matched individually according to age, sex, handedness, school education, as well as for IQ (estimated by the French National Adult Reading Test, fNART[62]). The protocol for human participants was approved by the Medical Ethical Review Committee of the Pitié-Salpêtrière Hospital (ID RCB n° 2012-A01460-43). All the participants gave their informed consent prior to the beginning of the study.

Fifty-two C57BL/6J male mice (26 *Sapap3*-null (KO) and 26 age matched wildtype (WT) littermates), 6-7 months old, were used. The mice were born, weaned (at postnatal day 21) and raised in the animal facility of the Brain and Spine Institute. Genotypes were determined by PCR of mouse tail DNA, using primer F1 (ATTGGTAGGCAATACCAACAGG) and R1 (GCAAAGGCTCTTCATATTGTTGG) for the

wildtype *Sapap3* allele (147 base pairs), and F1 and R2 (CTTTGTGGTTCTAAGTAC TGTGG; in neo cassette) for the mutant allele (222 base pairs). Before performing the task, they were living in group of 3-5 in ventilated cages with ad-libitum access to water and food, under a temperature of 20–22 °C and 50–60% humidity, and were maintained under a 12 h light/dark cycle (lights on from 8 a.m. to 8 p.m.). Mice started the task when they were at least 6 months old to maximize the chance to observe the grooming phenotype without severe skin lesions[18]. During the behavioral assessment, which approximately lasted a month, they were single-housed in the experimental cages with ad-libitum access to water under a 12 h light/dark cycle. The first 24-h in the experimental cage consisted in the habituation period. During this period, they had ad-libitum access to food and were video-taped from the top in order to quantify the self-grooming behavior (see "Grooming quantification" section). After this initial 24 h period, mice no longer had access to ad libitum food but a tablet was delivered each time they answered correctly during the behavioral task (see "Behavioral task" section). Their weight was monitored every day by the experimenter (less than 3 min handling), and they were supplemented with tablets if their weight went under 80% of their initial mass. During the entire protocol, mice were under continuous remote video-monitoring. Each animal experiment was approved by the Ethics committee Darwin/N°05 (Ministère de l'Enseignement Supérieur et de la Recherche, France) and conducted in agreement with institutional guidelines, in compliance with national and European laws and policies (Project n° 00659.01).

The detailed characteristics of the samples for both species are summarized in Table 3.

**Grooming quantification**. The recorded videos were manually analyzed using Kinovea v0.8.15. The self-grooming measures (number of grooming bouts and proportion of time spent grooming) were extracted from a 10-min activity period (a sufficient duration to highlight differences in the self-grooming behavior[63]) which started from 8 p.m. when mice become more active. When a mouse was not active at 8 p.m., the time window was moved forward until the mouse waked up and left the nest. When a mouse was already engaged in a grooming behavior at 8 p.m., the time window was moved forward to start after of the ongoing grooming sequence. If a mouse was still engaged in a grooming bout at the end of the time window, this one was moved forward in order to include only complete grooming bouts. Self-grooming was defined as one or more of the elements of the syntactic grooming chain in a flexible, non-chained order: elliptical strokes, small strokes, bilateral strokes, flank licks, and tail and genital licks[63–65]. Consistent with previous studies, we counted grooming bouts independently when they were separated by more than 2 s[63,66]. In addition, we considered two grooming bouts as independent when qualitatively different behavior interrupted the grooming sequence (i.e., jumping, locomotion, rearing)[63].

**Mouse automatized experimental chamber**. It has been shown that daily manipulation of animals in experimental procedures can increase stress and negatively impact behavioral results, including the assessment of behavioral flexibility[67]. To avoid this bias, especially in *Sapap3* KO mice, which express an anxious phenotype, we designed and used in our study an automated experimental chamber (Fig. 1a) where mice were exposed to the task 24 h a day. This behavioral apparatus consisted of a modified ENV-007CTX experimental chamber from Med Associates (Vermont, USA) with interior dimensions of 30.5 × 24.1 × 29.2 cm. The grid floor of the chamber was covered with a stainless-steel tray to receive bedding. On the left wall, two 2.8" TFT capacitive touchscreens (#2090, Adafruit, New York, USA) were placed symmetrically above the bedding tray. Each touchscreen was controlled by an Arduino (Leonardo model, Adafruit) interfaced with the I/O module (DIG-716B, Med Associates). On the right wall, a pellet dispenser placed in the center (ENV-203-20, Med Associates) was delivering 20 mg precision tablets (49.6% sucrose, 5TUL, Test Diet, Missouri, USA) into a pellet receptacle (ENV-303WX, Med Associates) equipped with an infrared head entry detector (ENV-303HDW, Med Associates). A water bottle (ENV-350RMX, Med Associates) was placed next to the pellet receptacle. On the ceiling, a micro camera (700TVL Super HAD CCD II with a 2.8 mm lens, Sony, Japan) associated with a red LED for night vision (5 mm, 55 cd) was fixed on the center and an aversive light (6 W LED spot) was vertically located above the pellet receptacle and tilted toward the touchscreens. The apparatus was controlled by Med-PC IV software (Med Associates) running on a desktop computer (under the Windows 7 OS) equipped with the DIG-700P2-R2 PCI interface card (Med Associates). The mouse lived in the experimental chamber for several weeks where water was provided ad libitum, and each trial could be self-initiated to get food. Therefore, these in-house experimental chambers allow a more naturalistic assessment of behavioral performance of the mice by respecting their nychthemeral rhythm and by avoiding prolonged and repetitive handling (less than 3 min per day for weighing only) and methodological constraints that could also affect animals behavior (such as food deprivation before the experiment)[68–70]. Moreover, our device allowed, as in humans, to repeat the measurement of interest through the collection of thousands of trials per mouse (Fig. 1b; see Supplementary Methods for details). To control for any environmental influence, the mice underwent the task in pairs (a KO mouse and its WT littermate). Prior to the beginning of the task itself, the mice were acclimatized to the experimental chamber for 24 h with a pellet (20 mg precision tablets containing 49.6% sucrose, 5TUL, Test Diet) delivered each time they poked their nose into the

food receptacle. After this acclimatization phase, the master program automatically triggered two successive pre-training instrumental phases (see Supplementary Methods for details) during which the animal learned to activate the screens and respond to visual stimuli. Finally, when the animal reached pre-defined criteria of completion of these pre-training phases, the reversal learning task could proceed (Supplementary Fig. S3).

**Reversal learning paradigm**. The human version of the reversal learning task (Fig. 1c, left) was administered in a computerized version adapted from Valerius et al.[71] and coded in MatLab R2013b (MathWorks) using the Psychophysics Toolbox v3 (http://psychtoolbox.org). The subjects sat in front of a 17" TFT monitor and a regular keypad. Two different abstract symbols from the Agathodaimon alphabet (white font on a black background) were randomly displayed on the left and right sides of the screen. Subjects had to choose one of these symbols by using either a left ("Q") or a right ("M") button-press according to the screen location of the stimulus. The subjects had the instruction to respond as fast as possible and the symbols remained on screen until their response. 750 ms after their response, a feedback (green or red smiley face displayed for 500 ms) indicated whether their answer was correct or not followed by the win or loss of one point, respectively. Inter-trials intervals were randomly sorted between 750 and 1250 ms. After 6 to 15 (randomized) consecutive correct responses, a reversal occurred and subjects had to adapt their response by selecting as correct the formerly wrong symbol. As classically done in human behavioral studies[72] to make reversal events less obvious, probabilistic errors were interspersed so that there was a 20% chance of receiving misleading feedback. In order to avoid successive probabilistic errors, we set a maximum of 3 possible continuous probabilistic errors and 3 possible probabilistic errors within a 10-trial sliding window. Moreover, probabilistic error never occurred on a reversal event and the 3 following trials. There were 3 breaks (every 6 reversal blocks) of up to 5 min and the pairs of symbols was changed after each break. The task ended after the completion of 20 reversals. All participants were first familiarized to the task with a few trials practice run (the training ended when the reversal criterion was reached).

The mouse version of the task (Fig. 1c right) was coded in MEDState Notation (MSN) language and was the same as the human version, with the feedback being deterministic as the main difference (i.e., there were no probabilistic error). As in the human task, the rewarded side was contingent on the stimuli presented on the screens to exclude any simple spatially-guided or even cue-guided strategy. When the mouse launched a trial by nose poking in the pellet receptacle, two distinct visual patterns of vertical and horizontal bars equally luminescent (Fig. 1c) were presented in white font on a black background on the left and right touchscreens with pseudo-randomized locations (within a 10-trial sliding window, the same pattern could not appear more than 3 times consecutively at the same location and more than a total of 7 times). The equiluminescent stimuli pair was chosen among those recommended by Horner et al.[73], i.e., grid and lines. Once a trial was initiated, the mouse had 60 s to respond before the screens turned off. If the mouse made a correct response, the two screens blinked and the mouse had 15 s to nose poke in the food receptacle for a reward. Otherwise, the aversive light was turned on for 5 s and the stimuli location remained the same for the subsequent trials until a correct response was made (corrective trials to avoid response lateralization). After completing a trial, the mouse could not launch another one within the next 5 s. When the mouse reached a criterion of 80% correct responses over the last 40 trials, a reversal occurred and the mouse had to choose the formerly wrong stimulus as the new correct response. The task ended after completion of 5 reversals, representing around 2000 trials performed over approximately 3 weeks (Fig. 1b and Supplementary Fig. S4). The first rewarding stimulus was counterbalanced between pairs.

For both species, the main behavioral measures to assess the subjects' performance were: the number of trials needed to reach the reversal criterion, and the reversal errors defined as the number of perseverative errors following a reversal event. For mice, this last measure was estimated as a proportion of errors in the so-called perseverative phase[74,75], defined as the block of consecutive trials following a reversal event until the performance rate reach 40%. These blocks were determined according to a change point analysis adapted from Gallistel et al.[76] (see "Change point analysis" section). Other parameters of interest, after the perseverative phase, were the probability of a spontaneous strategy change (SSC), i.e., switching to the unrewarded stimulus despite previous positive feedback; and the SSC errors, i.e., the number of consecutive errors after a spontaneous strategy change.

**Change point analysis**. To identify the perseverative phase in the mouse version of the task, a change point analysis was performed on the cumulative record of correct responses for each animal and for each reversal block[76]. This analysis allowed the detection of change points which marked significant variations in the slope of the cumulative record, a useful metric to identify changes in performance. We coded a recursive algorithm based on MatLab functions provided by Gallistel and colleagues[76] to search for putative change points in individual cumulative records of performance (i.e., the trial deviating maximally from a straight line drawn between the start of the record and the assessed point). A $\chi^2$ test was used to determine whether the frequencies of correct responses before and after the putative change

point significantly differed. A change point was retained if its logit value (log of the odds against the null hypothesis that there is no change) reached/exceeded the one defined by the user. We ran the algorithm on each reversal block for each animal starting with the highest (and very conservative) logit value of 6 and, as suggested by Rountree–Harrison et al.[77], counting down by increment of 0.1 until we could detect a change point or the lowest acceptable logit value of 1.3 was reached (no change point detected in this case). Therefore, a change point was marking a statistically significant distinction between the post reversal perseverative phase (performance level below 40%), and the learning phase (performance level exceeding 40%)[74,75].

**Statistics and reproducibility**. Bayesian statistics were used to overcome the multiple shortcomings of *null hypothesis significance testing*[78]. This approach allows us to assess not only the strength of the evidence against the null hypothesis but also the one in favor of it by computing the Bayes Factor[79] (BF), a ratio that contrasts, given the data, the likelihood of the alternative hypothesis ($H_1$) with the likelihood of the null hypothesis ($H_0$), hence the subscript $BF_{10}$. BF values have a natural and straightforward interpretation as indicative of "substantial" ($3 < BF_{10} < 10$), "strong" ($10 < BF_{10} < 30$), "very strong" ($30 < BF_{10} < 100$) and "decisive" ($BF_{10} > 100$) evidence in favor of $H_1$ (and conversely for $H_0$ for $BF_{10}$ values below $1/3$, $1/10$, $1/30$, and $1/100$, respectively). All the analyses were performed in JASP v0.9.2[80]. For group comparisons, two-tailed (or one-tailed when justified, indicated by $BF_{\pm0}$) Jeffreys–Zellner–Siow (JZS) paired t-tests[81] were carried out to analyze differences in continuous variables with an uninformed Cauchy prior ($\mu = 0$, $\sigma = 1/\sqrt{2}$), and two-tailed Gunel–Dickey (GD) contingency tables tests[82] for independent multinomial sampling were carried out to analyze differences in categorical variables with a default prior concentration of 1. For multiple-group comparisons, we used JZS ANOVA[83] with an uninformed multivariate Cauchy prior ($\mu = 0$, $\sigma = 1/2$) followed by post-hoc JZS t-tests and GD contingency tables tests for joint multinomial sampling. For all post-hoc analyses, we adjusted for multiplicity according to the Westfall approach[84], with the prior probability that the null hypothesis holds across all comparisons fixed to 0.5[85]. For the comparison of performance around reversal, a two-way mixed JZS ANOVA[83] including the within-subject factor "trial" (10 trials around reversal for humans and 100 for mice) and the between-subject factor "group" (OCD/KO vs. healthy/WT) were performed with an uninformed multivariate Cauchy prior ($\mu = 0$, $\sigma = 1/2$). When appropriate, effect sizes were reported, i.e., Cohen's *d* with its 95% Credible Interval (95CI) for JZS t-tests and the $\eta^2$ for JZS ANOVA.

To assess the inter-dependence between compulsive behavior severity and behavioral flexibility performance, we carried out correlation analyses between the severity score of the disorder and the behavioral parameters extracted from our task, as performed in previous studies in both species[33,71]. In humans, the different OCD clinical subtypes, measured by the dedicated OCI-R scale subscores, can have a distinct impact on behavioral flexibility[15]. Thus, we analyzed separately the relationship between the different OCD subtypes and behavioral flexibility. Likewise, we additionally explored the influence of depressive and anxious symptoms, as well as antidepressant dose on task parameters. These analyses relied on two-tailed JZS Pearson correlation tests[86] with an uninformed stretched β prior (width 1). The correlation coefficient *r* is reported along with its 95CI. The categorical variables were expressed as percentages and the continuous variables were expressed as means ± standard deviation (all the behavioral parameters were averaged out over the 5 reversals). The acquisition phase, defined as the trials needed to reach the reversal criterion for the first time, is thought to measure a baseline capacity for learning the associations, and not a reversal learning deficit[87]. It was therefore analyzed separately. All values were rounded to two decimal places.

**Cluster analysis**. To search for a potential subgroup of WT or KO mice which might behave differently in our task, we performed a two-step cluster analysis[88] using the four behavioral parameters extracted from our task (number of trials to reversal, reversal errors, SSC probability and SSC errors). This algorithm has the advantage of relying on the Bayes Information Criterion (BIC) to automatically determine the number of clusters; avoiding any subjective and biased selection[89]. The two-step cluster analysis procedure automatically selects the number of clusters minimizing the Bayes Information Criterion (BIC). The difference of the BIC values (noted ΔBIC) between the best solution and the alternative models is indicative of the strength of the evidence ($0 < |\Delta BIC| < 2$, weak evidence; $2 < |\Delta BIC| < 6$, positive evidence; $6 < |\Delta BIC| < 10$, strong evidence; and $|\Delta BIC| > 10$, very strong evidence)[89]. Two-step clustering also offers an overall goodness-of-fit measure called the silhouette measure which assess the quality of cluster separation. A silhouette measure of less than 0.20 indicates a poor solution quality, a measure between 0.20 and 0.50 a fair solution, whereas values of more than 0.50 indicate a good solution. Furthermore, the procedure indicates the relative importance of each variable in the determination of a specific cluster, with a value ranging from 0 (least important) to 1 (most important). Our clustering used log-likelihood as a measure of distance. It was followed by a stepwise discriminant analysis as a confirmatory procedure which included all mice (KO and WT). The analysis used the Wilks' lambda for variable selection and prior probabilities computed from group sizes along with the within-groups covariance matrix for classification. The Cohen's κ[90] was run to quantify the agreement between these two procedures (κ < 0.20 corresponding to a poor agreement; 0.21–0.40, a fair one; 0.41–0.60, a moderate one;

0.61–0.80, a good one; 0.81–1.00, a very good one)[91]. These analyses were performed in SPSS v25 (IBM) (See Supplementary Methods for more details).

**Reporting summary**. Further information on research design is available in the Nature Research Reporting Summary linked to this article.

## Data availability
The data that support the findings of this study are available from the corresponding author upon reasonable request. Source data for main Figs. 2 to 4 are provided with the publication: Supplementary data 1 (humans) and 2 (mice) for Fig. 2a; Supplementary data 3 (humans) and 4 (mice) for Figs. 2, 3, and 4.

## Code availability
The MatLab code for the human version (https://doi.org/10.5281/zenodo.4238706) and the MSN code for the mouse version (https://doi.org/10.5281/zenodo.4238732) of the task are available in GitHub repository under the GNU General Public License v3.0.

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

## Acknowledgements

This work was funded by *Agence Nationale de la Recherche–ANR-13-SAMA-0013-01_HYPSY* (N.B., L.M., E.B.); ANR program ERA-NET NEURON JTC 2013_TYMON (L.M., E.B.); *Investissements d'Avenir* program (Labex Biopsy) managed by ANR-11-IDEX-0004-02 (L.M., E.B.); and Program CARNOT Institute/ICM (E.B.). This work benefitted from the equipment and services of PHENO-ICMice and PRISME-ICM core facilities. The core facilities were supported by "Investissements d'avenir" (ANR-10-IAIHU-06 and ANR-11-INBS-0011-NeurATRIS) and "Fondation pour la Recherche Médicale". The authors thank Profs Guoping Feng and Ann Graybiel for generously providing the *Sapap3* KO mice. We thank Dr. Philippe Faure and Dr. Lizbeth Mondragon for their technical help in the design of the automated operant chambers. We also thank Christiane Schreiweis and Pauline Smith for the language and formatting improvements of the manuscript.

## Author contributions

N.B., K.N., L.M., and E.B. conceptualized the research; N.B., L.M., A.P. recruited the patients; N.B., K.N. coded the experimental tasks; N.B., E.B. designed the operant chambers; N.B. built the operant chambers, conducted experiments in both species, extracted the data and performed statistical analyses; N.B., K.N., E.B. interpreted the data; N.B., E.B. wrote the manuscript; K.N., L.M. edited the manuscript.

## Competing interests

The authors declare no competing interests.
