## [Peer Review File · Communications Biology]

Reviewers' comments:

Reviewer #1 (Remarks to the Author):

Benzina and colleagues take a strong translational approach to assess deficits in behavioral flexibility in obsessive-compulsive disorder (OCD) patients and in a mouse model of OCD. The major strength of this study is that the behavioral paradigm applied in the mouse model is comparable to the one applied in humans, as it shares the same sensory modality (visual). The study finds in humans and in mice that only subgroups of patients or OCD-like mice show a deficit in behavioral flexibility. The recognition of heterogeneity of OCD and also heterogeneity in the animal model is not completely novel, but highly relevant. More of such studies certainly advances the field towards more personalized diagnosis and treatment. It is also very commendable that the authors used an unbiased clustering method to search for subgroups in the mouse data set, instead of choosing an arbitrary cut-off. While the study is well designed and the findings are relevant for the field, several points should be improved as listed below.

1. Throughout the manuscript, the text should be revised for more concise and clearer writing (shorter sentences, remove any text which does not directly support the understanding and interpretation of the data).

2. The authors make a big point that their experimental design for the mouse study is advantageous over conventional rodent tests of behavioral flexibility, mostly since it is less stressful to the mouse. While I see some advantage in this paradigm ("no" handling, home-cage behavior, no food restriction), there are other stress factors related to this particular paradigm. The mice are isolated (singly-housed) for 3-4 weeks, which is certainly a big stressor for social animals. The mice only have access to food through the task, which might also be stressful especially after reversal of contingency. Furthermore, in the Supplementary Information, the authors mention that mice are weighted daily during the task, which is a manipulation by an experimenter (?), therefore the argument about "no handling" is not valid. It would probably be better to remove the point about handling / less stressful situation altogether as it is inconsistent and has no relevance for the finding.

3. As mentioned above in point 2, the authors emphasize that mice lived in the operant chambers without any human intervention or manipulation. However, the Supplementary Information states that mice were weighed daily, which is a daily manipulation by an experimenter. Please clarify this inconsistency.

4. Although statistically significant, the deficits of behavioral flexibility are minor (despite some deficits, mice and patients are able to relatively quickly adapt their behavior). This could be acknowledged/discussed in the text.

5. Introduction: Please describe better the previously published findings of behavioral flexibility in Sapap3-KO mice (study 26 and 27). Since behavioral flexibility relates to habitual behavior, it is also worth to touch base with the two Sapap3-KO studies that assessed habit formation (Hadjas 2019 and Ehmer 2020). While this part of the introduction should be expanded, the part of the introduction on brain circuits (line 66-72) can be more concise, because not directly relevant to the present data which are behavior only.

6. Please state in the main text method section what kind of food pellet the mice earned in the box (for example: "20 mg precision tablets containing 50% sucrose, 5TUL, Test Diet").

7. A better word for "ecological" (word used throughout the manuscript) would be "naturalistic".

8. The definition of "reversal errors" in mice is unclear (related to line 198).
9. Can the parameters SSC probability and SSC errors be defined in more details (line 202)? For example, what is the difference between an error and a strategy change? Although hinted to in line 203, the supplementary does not give more methodological details.
10. Throughout the manuscript: Why using the term "response lability" as a synonym of "spontaneous strategy change"? It would be easier for the reader to follow if the two terms were not used as synonyms. "Spontaneous strategy change" could be used only, since this term describes the behavior accurately, whereas "response lability" is somewhat generic.
11. Related to Figure 2: The authors used "above" or "up" and "below" or "down" to refer to panels in Figures. It is more common to use "top" and "bottom". At least the wording needs to be consistent between Figure legend and reference in the text.
12. Figure 4: Is the SCC probability shown here data extracted from the first reversal onwards, only from a reversal context (definition?), or does it include the acquisition phase? Please clarify.
13. Why do the "impaired" Sapap3-KO make fewer reversal errors? Would it not be expected that they make more reversal errors? Please explain.
14. Line 422: What is meant by "cognitive profile"? Maybe replace by a more appropriate term.
15. Discussion, line 425 onward: The referenced study (Manning et al.) shows hypoactivity in Sapap3-KO mice, but they also found that once the mice learned the instrumental task, hypoactivity does not interfere with the task. I therefore do not understand the argument in the discussion of the present manuscript and I do not see how this is relevant here. Please clarify or remove.

Reviewer #2 (Remarks to the Author):

This is a very interesting and well-conducted cross-species study, showing what appears to be an unexpected lability in decision-making choices in patients with OCD of checking sub-type that matches an apparently similar deficit in a sub-group of sapap3 ko mice. The statistical analysis is rather dense and the behavioral variables somewhat confusingly labelled in places, but the basic findings are clear. Of course, it would be more convincing if the sapap 3 ko mice "impaired" subgroup exhibited some other behavioral tendency that matched the checking in the subgroup of OCD patients- and also helped to explain the biological origin of their lability. The authors should also perhaps recognize that similar changes in cognitive flexibility in OCD in this paradigm have already been described in recent studies, as well as that decreased flexibility is much more convincingly shown in deterministic rather than probabilistic paradigms.

Specific points follow:

1. Although the behavioral parallels described between the mice and humans are clear, It is not so clear why a discriminant analysis is used to identify the relevant subgroups in mice, but not for the OCD patients?
2. Are the authors convinced that there is no effect of medication? The regression with fluoxetine equivalents may not have been significant but another comparison that perhaps could have been

made was between medicated and unmedicated patients.

3. It would be useful to compare the Sapap3 subgroups in terms of other measures (e.g. anxiety; elevated plus maze? and locomotor activity) as well as grooming and the rather indirect "activity" measure actually inferred in this ms. Are such measures available?

4. Have these OCD patients been studied using other tests of cognitive flexibility?

5. Other authors have already reported increased lability in OCD using very similar test paradigms and pointed out possible relationship with checking. The authors may have missed these recent publications, but they should probably be cited. Kanen et al (2019) *Psychopharmacology* (2019) 236:2337–2358 and Hauser et al *Psychol Med.* 47; 1246-1258. Both of these publications used formal reinforcement theory modelling to quantify these changes, finding notably reduced "stimulus stickiness" as a fundamental measure of reduced lability or "cognitive flexibility". There is also a relevant literature implicating changes in 5-HT manipulations in both humans and experimental animals- hence queries relating to medication, above. The Discussion might usefully discuss at last some of these findings.

Reviewer #3 (Remarks to the Author):

Benzina and colleagues investigated behavioral flexibility in OCD patients and Sapap3 KO mice. To be able to directly compare task performance of mice with that of humans, the authors designed a novel automated non-spatial reversal-learning task for mice and aligned it to a similar version for humans. Using this task in both species, the authors report considerable heterogeneity in behavioral flexibility in both species. Subsequently, they isolate sub-populations in both humans and mice, based on patient symptom profile and task performance, respectively. Based on this sub-population analysis, the authors report that compulsive(-like) behavior is not correlated with perseverative behavior in either species, but instead with response lability.

The topic of this study is of great interest to the fields of psychiatry and behavioral neuroscience, as studies investigating cognitive flexibility in OCD patients yielded inconsistent results in the past, which could be attributed to methodological differences, as well as clinical heterogeneity. Thus, this topic requires much more investigation. The authors conducted methodologically advanced experiments that deliver report-worthy findings. We would like to applaud the authors for investigating this matter in both humans and mice, as well as using a novel automated behavioral testing apparatus to avoid daily human handling of animals to reduce stress. Testing animal behavior in setups like this is highly advantageous. Furthermore, the overall experimental design and data presentation are sound, and the manuscript is well-written. However, there is a number of concerns, listed below, that should be addressed:

Major points

- 1) From a conceptual stand point, the use of Bayesian statistics is advantageous. However, it also brings along some practical difficulties. The authors tested many hypotheses in the same or dependent data sets. There needs to be a more detailed description how prior assessments were adjusted to account for multiple hypotheses, and how the authors dealt with this multiplicity issue in general.
- 2) Conceptually, considering the authors attempt to bring two species together to gain synergistic outcomes, the chosen approach leaves a gap. The authors identified relevant sub-populations in both humans and mice, based on patient symptom profile and task performance, respectively. In one case a cluster analysis was used, but not in the other.
- 2a) Why not perform a cluster analysis in humans, too?

2b) How much information is gained from performing a cluster analysis in mice that is based on data from a behavioral flexibility task to conclude that there is a sub-population that performs worse than another in this task?

2c) Why was the cluster analysis followed up with a discrimination analysis? If this was performed to support the cluster analysis, how was this support quantified and tested (other than qualitatively described)?

3) In how far do differences in task acquisition or attention contribute to behavioral inflexibility in the current paradigm?

4) The authors present a strong correlation between SSC probability with checking severity in OCD patients (and an absent correlation with SSC errors) in table 3. It would be interesting to perform the same analyses within Sapap3 KO mice and add to Table 3 or even Figure 3.

5) The authors mention that the subgroup-human analysis supports the result that checking symptoms were associated with a higher response lability (line 374-378). However, the BF10 value is only 1.19, which is not considered strong evidence. In addition, the next sentence suggests that OCD "checkers" were more labile than both healthy controls (I assume BF+0 should be BF10=3.51) and OCD "non-checkers" (again BF+0 should be BF10=2.22). Again, this would not be considered strong evidence. In Figure 4B, the comparison ncOCD vs cOCD suggests BF10>3, which is 2.22 according to the text. The authors should tone down these statements and change the sentences according to the BF10 values.

6) In the discussion, the authors compare their results with Manning et al. 2019 and state that, in the current study, they did not observe a decrease of activity of these mice. This data was not presented and needs to be added to the Supplementary Figures. In addition, I suggest to consider changing that sentence (line 423-433) into "...this result echoes what was reported in a recent study by Manning and colleagues".

7) The similar deficits in behavioral flexibility in subgroups of OCD and Sapap3 KO mice are discussed in the discussion. However, the authors should discuss the discrepancy between these groups as well, namely "checker" OCD patients show more errors than "impaired" KO show fewer errors compared to their respective controls.

Other points

1) Although the Supplementary Methods contain a lot of information, a number of key methodological details should be described in the methods section of the main text, including the type of pellets used, how animals initiated trials, and the relative timing of events during a trial. Likewise, it would clarify the manuscript if change point, spontaneous strategy change, and number of perseverative errors are explained in detail in the Methods section. If number of words is a concern, I suggest to direct parts of the detailed explanation of "Cluster analysis" to the supplements.

2) The results mention that only checking-symptom severity was related to an increased number of trials required to reach the reversal criterion (Figure 3A and Table 3) (line 303-305). Indeed, Figure 3A suggests that the correlation (trials to reversal vs checking severity) is BF10>3. However, table 3 shows that the BF10 for this correlation is 2.96.

3) Since this manuscript targets both audiences, clinical and animal researchers, it would be preferable if the authors explain abbreviations in more detail (e.g., in Table 1). Specifically, which questionnaires are related to which traits (compulsivity, anxiety, impulsivity, etc).

4) Legends of Tables 1, 2, and 3 state "A BF10 greater than one is in favor of a difference and vice versa". Although it is technically correct that a BF10 greater than one points towards H1, this sentence could be misleading. I recommend to mention that BF10 greater than 3 (or lower than 1/3) is considered strong evidence.

5) Given that the majority of "non-checker" OCD patients are female (no behavioral flexibility deficits), it would be interesting to statistically test if gender of OCD patients affects behavioral flexibility.

6) The authors mention that "weight was monitored every day" (supplementary methods). It is unclear such weighing was conducted by human experimenters or not. If yes, this should be stated,

since the automated experimental chamber is intended to avoid human contact.

7) It is unclear whether the Figure-1B example represents a wild type or Sapap3 KO mouse.

8) Consider adding acquisition data for "impaired" and "unimpaired" KO to Supplementary Figure S2.

9) The abstract begins with "The lack of behavioral flexibility has been proposed as one of the primary causes of compulsions". This statement should be backed up well with references in the introduction/discussion. Otherwise, it comes off as a straw man hypothesis.

10) The authors should change the description of the Sapap3 KO model from "predominant model" to "predominant genetic model". There is a number of other models presently used.

11) The authors should discuss whether fluoxetine treatment alters behavioral flexibility and whether that matters to this study.

12) Table 3's legend is insufficient.

13) The authors use the term "animal model of compulsive behavior" throughout the manuscript. I suggest to change this to "animal model for compulsive-like behavior" since this animal model is used to study behavioral phenomena modeling human behavior (but not implementing the same disorder).

Reviewer #5 (Remarks to the Author):

Although it makes sense to conceptualize compulsivity as impaired cognitive and behavioral flexibility, reversal learning studies so far showed inconsistent outcomes. Nevertheless, neuronal (OFC) reversal learning impairments are more consistently found in OCD patients and their relatives suggesting inflexibility is a useful neurobiological endophenotype to further investigate mechanistically in animal compulsivity models. The authors present an innovative and well-designed cross-species study using a comparable reversal learning tasks in human OCD patients and Sapap3 KO mice. Their study is important, first, in implicating the feasibility and validity of a cross-species compulsivity paradigm. In addition, their results show that a subgroup of compulsive (checking OCD) patients and mice are more behaviorally inflexible, however, being more response labile rather than perseverative. This could suggest a more insecure response pattern as an important cross-species mechanism of compulsivity. Some minor comments:

1. Please address the confounding factor that OCD patients were medicated with serotonergic, dopaminergic or GABA-ergic drugs and mice not.

2. Although it is great that the animals were tested in a prolonged ecological setting to minimize the influence of random stress and environmental confounds, wouldn't it have been more optimal to adopt something comparable (eg weekly repetitions) for the human testing given that human compulsivity is known to be highly responsive to situational stress?

3. Please explain more how to interpret the excessive response lability in a subtype of the Sapap3 KO mice. The results and author's interpretation suggest these mice behave more like OCD-checkers. However, these mice also display excessive washing/grooming which is more similar to the non-checking OCD-washers.

Referee expertise:

Referee #1: OCD, SAPAP3 KO model, neural circuits

Referee #2: behavioral analysis, computational modeling, cross-species analysis

Referee #3 & #4: (PI and trainee) OCD, behavioral flexibility, statistics

Reviewer #5: human studies, OCD, DBS, TMS

Reviewers' comments:

Reviewer #1 (Remarks to the Author):

Benzina and colleagues take a strong translational approach to assess deficits in behavioral flexibility in obsessive-compulsive disorder (OCD) patients and in a mouse model of OCD. The major strength of this study is that the behavioral paradigm applied in the mouse model is comparable to the one applied in humans, as it shares the same sensory modality (visual). The study finds in humans and in mice that only subgroups of patients or OCD-like mice show a deficit in behavioral flexibility. The recognition of heterogeneity of OCD and also heterogeneity in the animal model is not completely novel, but highly relevant. More of such studies certainly advances the field towards more personalized diagnosis and treatment. It is also very commendable that the authors used an unbiased clustering method to search for subgroups in the mouse data set, instead of choosing an arbitrary cut-off. While the study is well designed and the findings are relevant for the field, several points should be improved as listed below.

1. *Throughout the manuscript, the text should be revised for more concise and clearer writing (shorter sentences, remove any text which does not directly support the understanding and interpretation of the data).*

We understood that some key paragraphs of the original manuscript were lacking in clarity. We followed the reviewer's recommendations and substantially rewrote the manuscript to better highlight the outcomes.

2. *The authors make a big point that their experimental design for the mouse study is advantageous over conventional rodent tests of behavioral flexibility, mostly since it is less stressful to the mouse. While I see some advantage in this paradigm ("no" handling, home-cage behavior, no food restriction), there are other stress factors related to this particular paradigm. The mice are isolated (singly-housed) for 3-4 weeks, which is certainly a big stressor for social animals. The mice only have access to food through the task, which might also be stressful especially after reversal of contingency. Furthermore, in the Supplementary Information, the authors mention that mice are weighted daily during the task, which is a manipulation by an experimenter (?), therefore the argument about "no handling" is not valid. It would probably be better to remove the point about handling / less stressful situation altogether as it is inconsistent and has no relevance for the finding.*

We agree with the reviewer and acknowledge that the expression “no handling” was abusive since the mice were weighed once a day. Yet the handling was drastically reduced (less than 3 minutes/day) in our newly developed setup compared to conventional setup where the mice are moved back and forth from their homecage to the behavioural box every day. Importantly, in our protocol the weighting of the mice happened during the day at least 6 hours before they were actively engaged in the experiment since, as nocturnal animal, most of their trials were performed during the night.

Concerning the stress and anxiety potentially induced by the protocol in our setup, we agree that these results are not the primary focus of our current study and are not presented in details in the current manuscript. Therefore, we removed these statements as suggested by the reviewer. However, for information we have addressed these issues in a separate methodological publication (in preparation), which is specifically dedicated to describe and report on these newly developed automated behavioural chambers. In this separate work, we compared the effect of different protocols (automated vs daily manipulation) on the anxiety and motivation levels while the animal performed the same reversal learning task. Our results showed that our automated setup procedure did not induce additional chronic stress or lack of motivation (assessed by open-field and tail suspension tests) compared to conventional procedure with daily manipulation.

3. *As mentioned above in point 2, the authors emphasize that mice lived in the operant chambers without any human intervention or manipulation. However, the Supplementary Information states that mice were weighed daily, which is a daily manipulation by an experimenter. Please clarify this inconsistency.*

As explained in point 2, we acknowledge that there was a short human intervention (less than 3 minutes in total) for daily weighting done during daytime, i.e. several hours before the mouse was actively engaged in the task. Therefore, we modified the text in the main text line 139 to clarify this statement :

“Their weight was monitored every day by the experimenter (less than 3 minutes handling)...”

4. *Although statistically significant, the deficits of behavioural flexibility are minor (despite some deficits, mice and patients are able to relatively quickly adapt their behaviour). This could be acknowledged/discussed in the text.*

We completely agree with the reviewer and this statement was made in a paragraph of our supplementary material (see below). As suggested by the reviewer, we decided to move it to the main text in the discussion line 577:

“The use of Bayesian statistics is another strength of our study which allowed us to formally support the absence of prespecified differences, notably in terms of "classical" perseveration but also to quantify the weight of evidence in favour or against those between-group differences. This methodological strength also points out a limitation to the interpretation of our results, especially those obtained with humans. **Indeed, the differences highlighted between the checking patients and the healthy subjects are all supported by a Bayes Factor lower than 10 reflecting a substantial evidence but far from being decisive with a moderate effect size.** This speaks toward the need to replicate these results on a larger sample than the one included in this study.”

5. *Introduction: Please describe better the previously published findings of behavioral flexibility in Sapap3-KO mice (study 26 and 27). Since behavioral flexibility relates to*

habitual behavior, it is also worth to touch base with the two Sapap3-KO studies that assessed habit formation (Hadjas 2019 and Ehmer 2020). While this part of the introduction should be expanded, the part of the introduction on brain circuits (line 66-72) can be more concise, because not directly relevant to the present data which are behavior only.

As suggested by the reviewer we modified our introduction to better state what were the main results of the two studies published on behavioural flexibility in SAPAP3-KO animals line 75:

“Regarding behavioural flexibility, two recent studies^{22,23} have challenged this model in a spatial reversal learning task and found that *Sapap3* KO mice had impaired performances compared to controls after reversal event, although the type of deficit differed with increased perseveration found in one study²³ but not the other²². Interestingly, both studies found that these deficits were not correlated with the severity of compulsive-like grooming, suggesting that compulsivity and flexibility dimensions may be distinctly affected in this model. Moreover, one of the studies identified a lack of flexibility only in a subgroup of *Sapap3* KO mice²³. This result highlights inherent model heterogeneity, which echoes the clinical heterogeneity observed in OCD patients.”

We have also added and considered the studies on habit formation in the discussion of our current revised version line 548:

“Another dimension which could affect both compulsivity and flexibility is the overexpression of habitual behaviours. Some recent studies have favoured this idea in OCD patients⁸¹ and *Sapap3* KO mice⁸² but others dampened⁸³ or rejected this hypothesis⁸⁴. Thus, more evidence will be needed to fully understand the implication of habits in compulsive behaviours and flexibility.”

6. *Please state in the main text method section what kind of food pellet the mice earned in the box (for example: “20 mg precision tablets containing 50% sucrose, 5TUL, Test Diet”).*

We have moved this information from the supplementary material to the main text on line 207 as suggested by the reviewer:

“...with a pellet (20 mg precision tablets containing 49.6% sucrose, 5TUL, Test Diet)...”

7. *A better word for “ecological” (word used throughout the manuscript) would be “naturalistic”.*

We agree with the reviewer that this term is more appropriate, we did change it line 200: “Therefore, these in-house experimental chambers allow a more naturalistic assessment...”

8. *The definition of “reversal errors” in mice is unclear (related to line 198).*

We agree that the definition may have been misleading, we therefore change the sentence in the text line 266 as follow:

“For both species, the main behavioural measures to assess the subjects’ performance were: the number of trials needed to reach the reversal criterion, and **the reversal errors defined as the number of perseverative errors following a reversal event. For mice, this last measure was estimated as a proportion of errors in the so-called perseverative phase^{53,54}, defined as the block of consecutive trials following a reversal event until the performance rate reach 40%.** These blocks were determined according to a change point analysis adapted from Gallistel et al.⁵⁵ (see *Change point analysis*).”

9. *Can the parameters SSC probability and SSC errors be defined in more details (line 202)? For example, what is the difference between an error and a strategy change? Although hinted to in line 203, the supplementary does not give more methodological details.*

SSC and SSC errors are indeed errors but of different types. Overall, we have distinguished 3 different types of errors: the first type is those occurring just after a reversal event (namely, reversal errors), and the two other types are the error occurring outside of a reversal event (spontaneous strategy change, i.e. SSC, and their subsequent consecutive errors, i.e. SSC errors). To clarify this, we have explicitly stated what types of errors were SSC probability and SSC errors in the current version of the manuscript line 271:

“Other parameters of interest, after the perseverative phase, were **the probability of a spontaneous strategy change (SSC), i.e. switching to the unrewarded stimulus despite previous positive feedback; and the SSC errors, i.e. the number of consecutive errors after a spontaneous strategy change.**”

10. *Throughout the manuscript: Why using the term “response lability” as a synonym of “spontaneous strategy change”? It would be easier for the reader to follow if the two terms were not used as synonyms. “Spontaneous strategy change” could be used only, since this term describes the behaviour accurately, whereas “response lability” is somewhat generic.*

We have used “response lability” and its adjective form “labile” to improve readability while describing results and in the discussion. However, we agree it could be confusing as a generic term so we did define it clearly in the context of our study line 494:

“These results suggested that OCD “checkers” had a high response lability as identified through an elevated SSC probability.”

We carefully checked that it was used more appropriately in the current version.

11. *Related to Figure 2: The authors used “above” or “up” and “below” or “down” to refer to panels in Figures. It is more common to use “top” and “bottom”. At least the wording needs to be consistent between Figure legend and reference in the text.*

We thank the reviewer to have point this out and modified the wording in the text lines 348, 350, 372 and 373; and in the legend:

“Figure 2. Compulsivity is not related to behavioural flexibility.

(A) Changes in correct response probability around a reversal. **Top:** for humans with 10 trials around the reversal. Red line: OCD patients. Green line: healthy subjects. **Bottom:** for mice with 100 trials around the reversal. Red line: Sapap3 KO mice. Green line: WT mice. The data were smoothed using Savitzky-Golay algorithm for both species. **(B)** No difference was found between groups for both humans (left, n = 40 per group) and mice (right, n = 26 per group), neither when considering the number of trials needed to reach the reversal criterion (top) or the number of reversal errors (bottom). Triangle: group mean. Dot: individual mean. \emptyset : $BF_{10} < 1$. **(C) Top:** In OCD patients, the disease severity assessed by the Y-BOCS does not predict the number of trials needed to reach the reversal criterion. Dark line: linear fit. Gray area: confidence interval. **Bottom:** In Sapap3 KO mice, compulsive grooming severity assessed by the number of grooming bouts initiated (over a 10-minute period) does not predict the number of trials needed to reach the reversal criterion. Dark line: linear fit. Gray area: confidence interval.”

12. *Figure 4: Is the SCC probability shown here data extracted from the first reversal onwards, only from a reversal context (definition?), or does it include the acquisition phase? Please clarify.*

The acquisition phase, defined as the trials needed to reach the reversal criterion for the first time, is thought to measure a baseline capacity for learning associations, and not a reversal learning deficit (Patzelt et al., 2014). It was therefore analyzed separately. Thus, the SSC probability shown in our study is an average of the data extracted from the 5 post-reversal phases, and therefore does not include the data from the acquisition phase. We have now included this precision in the main text line 319:

“...(all the behavioural parameters were averaged out over the 5 reversals). The acquisition phase, defined as the trials needed to reach the reversal criterion for the first time, is thought to measure a baseline capacity for learning the associations, and not a reversal learning deficit⁶⁶. It was therefore analysed separately. “

13. *Why do the “impaired” Sapap3-KO make fewer reversal errors? Would it not be expected that they make more reversal errors? Please explain.*

Indeed, our first hypothesis was that the deficit of cognitive flexibility would be represented by a greater number of perseverative errors. However, and unexpectedly, we observed that the subgroup of Sapap3 KO impaired in our reversal task did not express a greater perseveration but a greater response lability. This higher response lability in Sapap3 KO mice can potentially reflect a conflict between opposite pre- and post-reversal stimulus-response associations in a context of uncertainty generated by the reversal. However, this surprising result requires further study to better understand the cognitive processes underlying this behaviour. We completed our discussion by referring to this result in a more explicit way on line 516:

“Additionally, we found in both species that, **contrary to what we would expect**, the deficit of behavioural flexibility observed in some subgroups was not underpinned by excessive perseverative behaviour after reversal but rather by greater response lability.”

14. *Line 422: What is meant by “cognitive profile”? Maybe replace by a more appropriate term.*

We clarified this expression by the term “neuropsychological profile” and better defined it line 524:

“the **neuropsychological profile** of checking patients is more disrupted than in other OCD patients, with major impairment in planning/problem solving, response inhibition and set-shifting (and therefore in executive functioning overall)⁷⁵.”

15. *Discussion, line 425 onward: The referenced study (Manning et al.) shows hypoactivity in Sapap3-KO mice, but they also found that once the mice learned the instrumental task, hypoactivity does not interfere with the task. I therefore do not understand the argument in the discussion of the present manuscript and I do not see how this is relevant here. Please clarify or remove.*

In our study, we did not observed deficit in locomotor activity or task engagement as in Manning et al. and van der Boom et al. studies. However, since the three studies (including

ours) used different experimental setup, tasks and protocols, it is difficult to conclude what could explain these discrepancies. Despite this difference, we agree that all the three studies conclude that locomotor activity could not explain the difference observed after reversal. Thus, we decided to remove this point which is somehow consensual across studies and is not relevant and necessary for the understanding of our work.

Reviewer #2 (Remarks to the Author):

This is a very interesting and well-conducted cross-species study, showing what appears to be an unexpected lability in decision-making choices in patients with OCD of checking sub-type that matches an apparently similar deficit in a sub-group of sapap3 KO mice. The statistical analysis is rather dense and the behavioral variables somewhat confusingly labelled in places, but the basic findings are clear. Of course, it would be more convincing if the sapap3 KO mice "impaired" subgroup exhibited some other behavioral tendency that matched the checking in the subgroup of OCD patients- and also helped to explain the biological origin of their lability. The authors should also perhaps recognize that similar changes in cognitive flexibility in OCD in this paradigm have already been described in recent studies, as well as that decreased flexibility is much more convincingly shown in deterministic rather than probabilistic paradigms.

Specific points follow:

- 1. Although the behavioral parallels described between the mice and humans are clear, It is not so clear why a discriminant analysis is used to identify the relevant subgroups in mice, but not for the OCD patients?*

Cluster analysis was performed in mice due to the absence of a behavioural phenotype allowing to distinguish one group of mice from another; this is in contrast to patients where OCD is not a homogeneous nosological entity with the existence of different clinical subtypes. Moreover, previous research has shown that these clinical subtypes are underpinned by cognitive impairments specific to each of them (Leopold and Backenstrass, 2015). Thus, as a first intention it was relevant to use these clinical subtypes as a basis to divide our group of patients; something impossible to achieve in our mice, where the heterogeneity of the group can only be revealed through the analysis of behavioural parameters. However, as judiciously suggested by the reviewer, we could confirm the pertinence of our results between species by conducting also clustering using behavioural parameters in our patients as we did with the mouse group. This analysis revealed an impaired subgroup of 7 patients representing 17.5% of all our patients; 85.7% of them in this subgroup being checkers (6 out of 7 patients). The fact that this subgroup consists almost exclusively of checkers reinforces the observation of an impairment specific to this clinical dimension we first used to distinguish the clinical subgroups. See supplementary results line 37.

- 2. Are the authors convinced that there is no effect of medication? The regression with fluoxetine equivalents may not have been significant but another comparison that perhaps could have been made was between medicated and unmedicated patients.*

As previously shown in the article, we did not observe influence of SSRI dose on performance, and no difference in the proportion of medicated vs unmedicated patients between the two subgroups. However, to definitely rule out the influence of treatment on performance in our task as recommended by the reviewer, we have performed a new analysis which shows no difference ($BF_{10} < 1$ for all parameters) between medicated patients ($n = 28$) and unmedicated patients ($n = 12$). This analysis has been added to the supplementary materials line 54.

3. *It would be useful to compare the Sapap 3 subgroups in terms of other measures (e.g. anxiety; elevated plus maze? and locomotor activity) as well as grooming and the rather indirect "activity" measure actually inferred in this ms. Are such measures available?*

Unfortunately, we do not have behavioural measures other than those presented in the manuscript. However, at your suggestion, we have quantified locomotor activity on a subgroup of 10 mice (5 wild type and 5 Sapap3 KO) to confirm that our measure of activity represented by the number of trials initiated per hour is a good proxy for locomotor activity. To do so, we used the same methodology as described for the quantification of grooming with the use of semi-automatic video tracking. We found a positive correlation between these two types of measurements ($BF_{+0} = 3.96$, $r = 0.63$ [0.08 0.87]), confirming that the frequency of trials initiated is well representative of the locomotor activity.

4. *Have these OCD patients been studied using other tests of cognitive flexibility?*

It would have been indeed interesting to have more test to assess cognitive flexibility but we had access to each patient for a limited amount of time and we could only perform an extensive clinical assessment and run our behavioural reversal task in two distinct sessions. Unfortunately, we could not perform further experimental procedures.

5. *Other authors have already reported increased lability in OCD using very similar test paradigms and pointed out possible relationship with checking. The authors may have missed these recent publications, but they should probably be cited. Kanen et al (2019) Psychopharmacology (2019) 236:2337–2358 and Hauser et al Psychol Med. 47; 1246-1258. Both of these publications used formal reinforcement theory modelling to quantify these changes, finding notably reduced "stimulus stickiness" as a fundamental measure*

of reduced lability or "cognitive flexibility". There is also a relevant literature implicating changes in 5-HT manipulations in both humans and experimental animals- hence queries relating to medication, above. The Discussion might usefully discuss at last some of these findings.

We thank the reviewer for this comment and acknowledge that we should better cite these recent works which, as in our study, propose that response lability could be one of the main dimension affected in OCD patients (or at least in subgroup of patients). Kanen et al., (2019) reported reduced stimulus stickiness in OCD patients and Hauser et al., (2017) show that OCD patients would have a lower probability of win-stay compared to control subjects with a lower "perseveration" parameter in their model. These studies echo our results indeed, while we confirm and extend these results by taking into account the different subtypes of OCD. As recommended, we discuss these studies in the discussion part of the current manuscript line 530:

"The excessive response lability observed in some subgroups for both species echoes, particularly in humans, the results of computational studies performed by Kanen and colleagues⁷⁶ that reported reduced stimulus stickiness in OCD patients and by Hauser and colleagues⁷⁷ that showed a lower win-stay probability in OCD patients compared to control subjects with a lower "perseveration" parameter in their model. However, unlike our study, the different OCD subtypes were not taken into account and therefore the influence of the "checking" dimension in their result cannot be excluded."

Reviewer #3 (Remarks to the Author):

Benzina and colleagues investigated behavioral flexibility in OCD patients and Sapap3 KO mice. To be able to directly compare task performance of mice with that of humans, the authors designed a novel automated non-spatial reversal-learning task for mice and aligned it to a similar version for humans. Using this task in both species, the authors report considerable heterogeneity in behavioral flexibility in both species. Subsequently, they isolate sub-populations in both humans and mice, based on patient symptom profile and task performance, respectively. Based on this sub-population analysis, the authors report that compulsive(-like) behavior is not correlated with perseverative behavior in either species, but instead with response lability.

The topic of this study is of great interest to the fields of psychiatry and behavioral neuroscience, as studies investigating cognitive flexibility in OCD patients yielded inconsistent results in the past, which could be attributed to methodological differences, as well as clinical heterogeneity. Thus, this topic requires much more investigation. The authors conducted methodologically advanced experiments that deliver report-worthy findings. We would like to applaud the authors for investigating this matter in both humans and mice, as well as using a novel automated behavioral testing apparatus to avoid daily human handling of animals to reduce stress. Testing animal behavior in setups like this is highly advantageous. Furthermore, the overall experimental design and data presentation are sound, and the manuscript is well-written. However, there is a number of concerns, listed below, that should be addressed:

Major points

- 1. From a conceptual stand point, the use of Bayesian statistics is advantageous. However, it also brings along some practical difficulties. The authors tested many hypotheses in the*

same or dependent data sets. There needs to be a more detailed description how prior assessments were adjusted to account for multiple hypotheses, and how the authors dealt with this multiplicity issue in general.

Contrary to frequentist statistics where multiple comparisons need to be corrected at the risk of increasing type I error with the number of comparisons performed; the Bayesian approach is only slightly affected by this problem. Indeed, it is not necessary to make any adjustment for multiplicity in the Bayesian framework (Dienes, 2011; Gelman, 2016; Harrell Jr, 2019) because it is automatically controlled by the priors, provided they are not noninformative (uniform/flat priors) (Gelman, 2016). Indeed, the use of noninformative priors leads to the same problem as with frequentist statistics: a type I error risk inflation (Lemoine, 2019). On the contrary, the use of weakly informative default priors such as Cauchy's distribution for example allows, by integrating a minimum of available knowledge (in the case of the default Cauchy's prior we used, that in cognitive neuroscience small effect sizes are more likely than large effect sizes), to control multiplicity by shrinking the parameter estimates towards zero (Gelman and Tuerlinckx, 2000; Lemoine, 2019; Neath et al., 2018), thus making data from small samples and noisy data unlikely to lead to a conclusion; low powered data resulting in larger shrinkage (Lemoine, 2019). This makes Bayesian analyses more conservative and much less likely to lead to confident erroneous conclusions (Gelman and Tuerlinckx, 2000). However, although some authors rely on this property to argue that multiplicity is not a problem even in post-hoc analyses following an ANOVA for example (Lemoine, 2019), the interdependence of these types of tests cannot be ignored; the results of one test are necessarily informative on the other two (in the case of 3 subgroups). Thus, for all our subgroup analyses, we adjusted for multiplicity according to the Westfall approach (Westfall, 1997). More precisely, the prior probability that the null hypothesis holds across all comparisons was fixed to 0.5 (de Jong, 2019). We added this information in the main text line 303:

“For all post-hoc analyses, we adjusted for multiplicity according to the Westfall approach⁶³, with the prior probability that the null hypothesis holds across all comparisons fixed to 0.5⁶⁴.”

2. Conceptually, considering the authors attempt to bring two species together to gain synergistic outcomes, the chosen approach leaves a gap. The authors identified relevant sub-populations in both humans and mice, based on patient symptom profile and task performance, respectively. In one case a cluster analysis was used, but not in the other.

a. Why not perform a cluster analysis in humans, too?

As responded to reviewer #1, a cluster analysis was performed in mice due to the absence of a behavioural phenotype allowing to distinguish one group of mice from another; this is in contrast to patients where OCD is not a homogeneous nosological entity with the existence of different clinical subtypes. Moreover, previous research has shown that these clinical subtypes are underpinned by cognitive impairments specific to each of them (Leopold and Backenstrass, 2015). Thus, it was relevant to use these subtypes as a basis for dividing our group of patients; something impossible to achieve in our mice, where the heterogeneity of the group can only be revealed through the analysis of behavioural parameters. However, we have taken this remark into account by conducting a clustering using behavioural parameters in our patients as we did with the mouse group (see supplementary results line 37). This analysis revealed an impaired subgroup of 7 patients representing 17.5% of all our patients;

85.7% of them being checkers (6 out of 7 patients). The fact that this subgroup consists almost exclusively of checker patients reinforces the observation of an impairment specific to this clinical dimension. We added this precision in the main text line 434:

“Noteworthy, we conducted the same clustering procedure on the human data with comparable results than the ones observed in mice. Two clusters were found in OCD patients (with one cluster constituted predominantly of “OCD “checkers”) and only one for healthy controls (see Supplementary Results for details).”

- b. How much information is gained from performing a cluster analysis in mice that is based on data from a behavioral flexibility task to conclude that there is a sub-population that performs worse than another in this task?*

As explain in the material and methods section, the two-step cluster analysis automatically selects the number of clusters minimizing the Bayes Information Criterion (BIC). Thus, if the data had not favoured clustering, the algorithm would have retained a more parsimonious single-cluster model. In our case, the model with 2 sub-populations of mice differing in their performance in our task was automatically retained as the optimal solution for the data with a BIC of 92.82 (96.13 for no sub-population and 109.66 for 3 sub-populations). This $\Delta\text{BIC} > 3$ difference in favour of the 2-cluster model is indicative of “positive” evidence according to the interpretation proposed by (Raftery, 1995) for whom a ΔBIC below 2 corresponds to “weak” evidence; between 2 and 6, is “positive”, between 6 and 10 is “strong”, and above 10 is very strong evidence. We added these details in the main text line 399:

“The resulting ΔBIC of -3.31 indicated a positive evidence in favour of this clustering.”

- c. Why was the cluster analysis followed up with a discrimination analysis? If this was performed to support the cluster analysis, how was this support quantified and tested (other than qualitatively described)?*

Indeed, the discriminant analysis was performed to support the cluster analysis. Thus, Cohen's κ (Cohen, 1960) was run to quantify the agreement between the two classifications. This agreement was moderate with $\kappa = .528$ (95% CI, .324 to .732), $p < .0005$ (Interpretation of κ according to Landis and Koch (1977): $\kappa < 0.20$ corresponding to a poor agreement; 0.21-0.40, a fair one; 0.41-0.60, a moderate one; 0.61-0.80, a good one; 0.81-1.00, a very good one). This moderate agreement can be explained by the fact that WT and unimpaired KO mice overlap in their task performance. We added this information in the main text line 428:

“These results were consistent with those of the two-step cluster analysis: the “unimpaired” *Sapap3* KO mice cluster was closer to the WT mice cluster (Figure 3C) with a moderate agreement between the two analyses ($\kappa = 0.53$, 95% CI: [0.32 0.73], $p < .0005$).”

- 3. In how far do differences in task acquisition or attention contribute to behavioral inflexibility in the current paradigm?*

According to our results, the acquisition phase does not influence post-reversal performance with no difference in terms of acquisition between impaired KO and WT mice, whereas these mice differ in terms of post-reversal performance. In humans, although patients differ in terms of acquisition, this latter does not correlate with the number of post-reversal trials, which is supportive of the lack of contribution of this parameter to behavioural inflexibility in our paradigm. Since an alteration in attention should logically lead to an impairment in task

acquisition, we can indirectly assume that it has little influence on post-reversal performance. However, confirming this assumption would have required using a separate attentional task and correlating its outcome with our main task's results, which was impossible to accomplish in the limited time we had access to the patients. We therefore agree that this is a limitation of the interpretation of our results and we acknowledge it in the discussion line 551:

“Finally, it cannot be ruled out that the deficit found is linked to an impairment that is not task-specific, such as an attentional impairment, whether primary⁷⁵ or secondary to obsessive activity in patients (particularly for checking patients, as the reversal increases uncertainty and thus possibly the obsessive doubt)⁸⁵. It would thus be important in the future to simultaneously assess attention, like other cognitive dimensions, to better characterize the underlying mechanisms of this observed behavioural inflexibility.”

4. *The authors present a strong correlation between SSC probability with checking severity in OCD patients (and an absent correlation with SSC errors) in table 3. It would be interesting to perform the same analyses within Sapap3 KO mice and add to Table 3 or even Figure 3.*

We agree with the reviewer that an analysis between checking severity and SSC probability also performed in mice would add to the translational validity of our study. However, in mice we did not had access to this dimension so we cannot perform the suggested analysis despite its obvious interest. To answer this suggestion, in the future one could try to specifically assess the checking dimension in the Sapap3 KO mice by developing a dedicated task as it was done with rat (Vousden et al., 2020). Having a quantitative assessment of checking severity in the animals would allow us to correlate it with the SSC probability (though it would mean that two tasks have to be run in the same animals). We discussed this point in the discussion part of the main text line 544:

“Obviously, in mice these checking behaviours are not directly observable, but it would be interesting to test if uncertainty monitoring and checking behaviours are also affected in *Sapap3* KO mice. Indeed, one could expect that abnormal increase of uncertainty after reversal would provoke the excessive lability of their behaviour (e.g. with mice over-checking if the previously rewarded stimulus is still valid).”

5. *The authors mention that the subgroup-human analysis supports the result that checking symptoms were associated with a higher response lability (line 374-378). However, the BF10 value is only 1.19, which is not considered strong evidence. In addition, the next sentence suggests that OCD “checkers” were more labile than both healthy controls (I assume BF+0 should be BF10=3.51) and OCD “non-checkers” (again BF+0 should be BF10=2.22). Again, this would not be considered strong evidence. In Figure 4B, the comparison ncOCD vs cOCD suggests BF10>3, which is 2.22 according to the text. The authors should tone down these statements and change the sentences according to the BF10 values.*

We completely agree with the reviewer and we did our best to appropriately qualify the strength of the evidence regarding this specific result. To avoid any ambiguity on the interpretation of our results, we have now added to the main text a paragraph discussing this aspect (previously in the supplementary discussion), which states that a low to moderate level of evidence for human results indeed requires replication line 579:

“This methodological strength also points out a limitation to the interpretation of our results, especially those obtained with humans. Indeed, the differences highlighted between the checking patients and the healthy subjects are all supported by a Bayes Factor lower than 10 reflecting a low to substantial evidence but far from being decisive with a moderate effect size. This speaks toward the need to replicate these results on a larger sample than the one included in this study.”

Typographically we used the label “BF+0”, to be explicit about the fact that this test was one-sided (because of the a priori on the direction of the difference). Concerning figure 4, we thank the reviewer for pointing out this error which has been corrected to match the BF value :

Figure 4. An excessive response liability underlies the reversal learning deficit.

(A) Deficits of behavioural flexibility found in both OCD “checkers” patients and “impaired” KO mice subgroups were not explained by a greater perseveration (in term of reversal errors). **(B)** Instead, both OCD “checkers” and “impaired” KO mice showed an increased SSC probability compared to other groups, suggesting a higher response liability. *ncOCD*: “non-checkers”. *cOCD*: “checkers”. *uKO*: “unimpaired” KO mice. *iKO*: “impaired” KO mice. Triangle: group mean. Dot: individual mean. \emptyset : $BF_{10} < 1$. \ast : $BF_{10} < 3$. $\ast\ast$: $BF_{10} \geq 3$. $\ast\ast\ast$: $BF_{10} \geq 10$. $\ast\ast\ast\ast$: $BF_{10} \geq 30$. $\ast\ast\ast\ast\ast$: $BF_{10} \geq 100$.

6. In the discussion, the authors compare their results with Manning et al. 2019 and state that, in the current study, they did not observe a decrease of activity of these mice. This data was not presented and needs to be added to the Supplementary Figures.

These data are available in supplementary figure S3, although this is indeed an indirect measure (number of trials per hour and in the first and last 48 hours).

In addition, I suggest to consider changing that sentence (line 423-433) into “...this result echoes what was reported in a recent study by Manning and colleagues”.

We have made the suggested modification line 526:

“...the identification of only a subgroup of *Sapap3* KO mice impaired in our reversal learning task echoes what was reported in a recent study by Manning and colleagues²³...”

7. *The similar deficits in behavioral flexibility in subgroups of OCD and Sapap3 KO mice are discussed in the discussion. However, the authors should discuss the discrepancy between these groups as well, namely “checker” OCD patients show more errors than “impaired” KO show fewer errors compared to their respective controls.*

We assume that the reviewer is referring to SSC errors and reversal errors. In humans, there is no statistical difference for both error types. In mice, however, there is a significant one. This is consistent with an increased SSC probability and therefore a lower tendency to persevere. The difference between the two species could be explained by the fact that the human version of the task was probabilistic and thus a negative feedback could be interpreted as a probabilistic error and therefore not lead to a switch. We discuss this discrepancy as suggested by the reviewer on line 99 of the supplementary information:

“Nevertheless, this difference could explain some difference observed in our results, such as the lower tendency to persevere of the “impaired” *Sapap3* KO mice compared to control, not observed in the human version of the task with OCD checkers. The difference between the two species could be explained by the fact that the human version of the task was probabilistic and thus a negative feedback could be interpreted as a probabilistic error and therefore not lead to a switch in their response; patients not having an increased sensitivity to probabilistic negative feedback.”

Other points

1. *Although the Supplementary Methods contain a lot of information, a number of key methodological details should be described in the methods section of the main text, including the type of pellets used, how animals initiated trials, and the relative timing of events during a trail. Likewise, it would clarify the manuscript if change point, spontaneous strategy change, and number of perseverative errors are explained in detail in the Methods section. If number of words is a concern, I suggest to direct parts of the detailed explanation of “Cluster analysis” to the supplements.*

As suggested by the reviewer and to clarify our methodological approach, we have indeed moved a substantial part of the supplementary material to the main text of the current revised manuscript.

2. *The results mention that only checking-symptom severity was related to an increased number of trials required to reach the reversal criterion (Figure 3A and Table 3) (line 303-305). Indeed, Figure 3A suggests that the correlation (trials to reversal vs checking severity) is $BF_{10} > 3$. However, table 3 shows that the BF_{10} for this correlation is 2.96.*

We apologize for this mistake that we have corrected it in the current version of the manuscript:

Figure 3. Only a subgroup of OCD patients and *Sapap3* KO mice is impaired.
(A) Only the severity of the checking symptoms (measured by the OCI-R checking subscore) predicts the number of trials needed to reach the reversal criterion. The higher the checking symptoms severity, the higher the number of trials. *Dark line: linear fit. Gray area: confidence interval.* **(B)** OCD patients with predominant checking symptoms (“checkers” subgroup) were segregated from the others (“non-checkers” and healthy controls subgroups). Only OCD “checkers” patients were impaired in terms of number of trials needed to reach the reversal criterion. *ncOCD: “non-checkers”. cOCD: “checkers”. Triangle: group mean. Dot: individual mean.* **(C)** A two-step cluster analysis using four behavioural parameters (number of trials to reversal, reversal errors, SSC probability and SSC errors) found two distinct clusters within the *Sapap3* KO mice and was confirmed by a stepwise discriminant analysis. *The intersection point of the lines indicates the group’s centroid.* **(D)** Only the “impaired” KO mice needed more trials to reach the reversal criterion compared to the other “unimpaired” KO and WT mice. *uKO: “unimpaired” KO mice. iKO: “impaired” KO mice. Triangle: group mean. Dot: individual mean.*
 \emptyset : $BF_{10} < 1$. \circ : $BF_{10} < 3$. *: $BF_{10} \geq 3$. **: $BF_{10} \geq 10$. ***: $BF_{10} \geq 30$. ****: $BF_{10} \geq 100$.

3. Since this manuscript targets both audiences, clinical and animal researchers, it would be preferable if the authors explain abbreviations in more detail (e.g., in Table 1). Specifically, which questionnaires are related to which traits (compulsivity, anxiety, impulsivity, etc).

We agree with the reviewer that the results and wording of our translational study needs to be accessible for both audiences. Therefore, we have made the suggested clarifications which will be useful for non-clinicians line 106:

“To assess severity and clinical subtypes of obsessive-compulsive (OC) symptoms, the Yale-Brown Obsessive-Compulsive Scale (YBOCS³⁵) was administered only for OCD patients, and the Obsessive Compulsive Inventory Revised (OCI-R³⁶) was used to measure all participants’ OC characteristics.” And line 115:

“Depression, trait/state anxiety and impulsivity were assessed respectively with the short version of the Beck Depression Inventory (BDI³⁷), the Spielberger’s State-Trait Anxiety Inventory (STAI³⁸) and the Barratt impulsivity scale (BIS-10³⁹) in their French version.”

We also added these clarifications to the legend of the table 1:

“Table 1. Demographic and clinical characteristics of the samples.

Mean values (with standard deviation) of demographic and clinical characteristics for human and mice populations. A BF_{10} greater than one is in favour of a difference (and vice versa). The further the BF_{10} is from one, the greater the evidence (and vice versa). A BF_{10} greater than 3 (or lower than Y_4) is commonly considered a significant evidence. The * indicates a JZS ANOVA BF_{10} with the results of post hoc tests given after the BF_{10} value; “=” indicating an absence of difference. AAP: atypical antipsychotic. OCI-R : Obsessive Compulsive Inventory Revised. YBOCS : Yale-Brown Obsessive-Compulsive Scale. BDI : Beck Depression Inventory. STAI : Spielberger's State (A) – Trait (B) Anxiety Inventory. BIS-10 : Barratt impulsivity scale. fNART : French National Adult Reading Test.”

4. Legends of Tables 1, 2, and 3 state “A BF_{10} greater than one is in favor of a difference and vice versa”. Although it is technically correct that a BF_{10} greater than one points towards H_1 , this sentence could be misleading. I recommend to mention that BF_{10} greater than 3 (or lower than $1/3$) is considered strong evidence.

We agree with the reviewer and we have modified the legend of these tables according to this suggestion:

“Table 1. Demographic and clinical characteristics of the samples.

Mean values (with standard deviation) of demographic and clinical characteristics for human and mice populations. A BF_{10} greater than one is in favour of a difference (and vice versa). The further the BF_{10} is from one, the greater the evidence (and vice versa). A BF_{10} greater than 3 (or lower than Y_4) is commonly considered a significant evidence. The * indicates a JZS ANOVA BF_{10} with the results of post hoc tests given after the BF_{10} value; “=” indicating an absence of difference. AAP: atypical antipsychotic. OCI-R : Obsessive Compulsive Inventory Revised. YBOCS : Yale-Brown Obsessive-Compulsive Scale. BDI : Beck Depression Inventory. STAI : Spielberger's State (A) – Trait (B) Anxiety Inventory. BIS-10 : Barratt impulsivity scale. fNART : French National Adult Reading Test.”

“Table 2. Behavioural parameters.

Mean values (with standard deviation) of main behavioural parameters of the reversal learning task in both species. A BF_{10} greater than one is in favour of a difference and vice versa. The further the BF_{10} is from one, the greater the evidence. A BF_{10} greater than 3 (or lower than Y_4) is commonly considered a significant evidence. For JZS ANOVA, the results of post hoc tests are given after the BF_{10} value with “=” indicating an absence of difference. SSC : Spontaneous Strategy Change. SCAPE : Strategy Change After a Probabilistic Error.”

“Table 3. Correlations between clinical and task parameters.

In patients, only the checking dimension influences the performance in our task; whereas in *Sapap3* KO mice, there is no influence of grooming severity. A BF_{10} greater than one is in favour of a correlation and vice versa. The further the BF_{10} is from one, the greater the evidence. A BF_{10} greater than 3 (or lower than Y_4) is commonly considered a significant evidence. OCI-R : Obsessive Compulsive Inventory Revised. YBOCS : Yale-Brown Obsessive-Compulsive Scale. BDI : Beck Depression Inventory. STAI : Spielberger's State (A) – Trait (B) Anxiety Inventory. BIS-10 : Barratt impulsivity scale. fNART : French National Adult Reading Test. SSC : Spontaneous Strategy Change. SCAPE : Strategy Change After a Probabilistic Error.”

5. Given that the majority of “non-checker” OCD patients are female (no behavioral flexibility deficits), it would be interesting to statistically test if gender of OCD patients affects behavioral flexibility.

Although the number of women in the “non-checkers” group is higher than in the “checkers” group, this difference is not statistically apparent as it is shown in Table 1. Nevertheless, we agree with the reviewer and we have now included the gender factor in our analyses. As a result, we observed that the gender had no influence on performance in our task. These new analyses have been added in the supplementary materials (supplementary tables S4 and S6) and in the main text line 461 :

“Comorbid anxiety disorder or gender effect were absent in humans (Supplementary Table S3 and S4).”

And line 500 :

“no comorbid anxiety disorder or gender effect was found (Supplementary Table S5 and S6).”

6. The authors mention that “weight was monitored every day” (supplementary methods). It is unclear such weighing was conducted by human experimenters or not. If yes, this should be stated, since the automated experimental chamber is intended to avoid human contact.

As suggested, we have tempered the “daily manipulation avoidance” statement by specifying in the main text that the mice are weighed once a day with a very brief manipulation at this precise moment, line 139:

“...monitored every day by the experimenter (less than 3 minutes handling)...”

7. It is unclear whether the Figure-1B example represents a wild type or Sapap3 KO mouse.

We apologize for having omitted this information. This figure indeed represents a wildtype animal and this is now explicitly stated in the figure’s legend:

“Figure 1. The reversal learning task.

(A) Illustration of the behavioural apparatus. Up to 8 operant conditioning chambers run in parallel with mice living and working with minimal human intervention. Each operant chambers was equipped with capacitive touch-sensitive screens (left), pellet and water dispenser (right), and LED lights (top). (B) Example of one WT mouse performance (smoothed over a 40-trial sliding window) across five reversals. (C) Design of the human (left) and the mouse (right) versions of the reversal learning task. On each trial, the subject had to make a choice between two different stimuli displayed on the screens. Depending on their choice, positive (for correct response) or negative (for incorrect response) feedback was provided. When the subject had learned the correct association, the reward contingencies were reversed without notice (see Material and Methods for details).”

8. Consider adding acquisition data for “impaired” and “unimpaired” KO to Supplementary Figure S2.

Figure S2 only illustrates the duration of the experiment with our system by taking the example of wildtype mice so that the reader has an idea of the duration of each stage of the task. At this stage of the manuscript the “impaired” and “unimpaired” groups are not defined so we prefer not to supercharge this figure with this mention to avoid any confusion. However, the data required by the reviewer could be found in Table 2 and we have added the Sapap3 KO mice data to this figure:

Supplementary Figure S2. Average number of days required to complete the different phases of the task. $n = 26$ per group. Bar: median. Circles: individual data.

9. *The abstract begins with “The lack of behavioral flexibility has been proposed as one of the primary causes of compulsions”. This statement should be backed up well with references in the introduction/discussion. Otherwise, it comes off as a straw man hypothesis.*

As the reviewer indicates, it is true that the wording of the sentence makes a strong statement. We propose to replace "primary" by "underlying"; bearing in mind that it is clearly stated that it could be *one of the* causes and not *the* cause. This relation between the two dimensions has been suggested in several studies (Gruner and Pittenger, 2017) that we have cited in the introduction line 62:

“Thus, behavioural flexibility impairments have been proposed has one of the causes of compulsive behaviours².”

10. *The authors should change the description of the Sapap3 KO model from “predominant model” to “predominant genetic model”. There is a number of other models presently used.*

We have changed the text accordingly on line 69:

“...the current predominant genetic model of compulsive-like behaviour¹⁸.”

11. *The authors should discuss whether fluoxetine treatment alters behavioural flexibility and whether that matters to this study.*

This remark by the reviewer is indeed very important and should be better discussed in our manuscript. Indeed, we have shown the absence of influence of SSRIs on performance in our task both by correlation analysis and by comparison of treated and untreated patients (data added in supplementary results line 54. Furthermore, it has been shown in rodents that chronic administration of an SSRI has a positive impact on performance by reducing the number of perseverative errors and promoting a win-stay strategy (Bari et al., 2010). These results in both species supports the idea that serotonergic treatments do not influence the results found in behavioural flexibility, and more particularly in our task. We have then discussed more in details these results line 557:

“A potential limitation of our study that can be pointed out is the difference in terms of medication between the two species. Indeed, while our *Sapap3* KO mice were free of any pharmacological treatment that could alter their performance, OCD patients were largely on serotonergic treatment. It is thus logical to think that the results obtained may reflect the effect of the treatment and not their disorder. However, we were able to show the absence of influence of serotonergic medication on performance in our task with no difference between medicated and unmedicated patients. Further, it has been shown that chronic administration of a selective serotonin reuptake inhibitor reduces perseveration and promotes a win-stay strategy in this type of task⁸⁶, which is contrary to our results. We can therefore be confident of the cross-species validity of these findings.”

12. *Table 3’s legend is insufficient.*

We added the precision concerning the interpretation of the BF and indicated that this table concerns the influence of different clinical parameters on the performance in our task:

“**Table 3. Correlations between clinical and task parameters.**”

In patients, only the checking dimension influences the performance in our task; whereas in *Sapap3* KO mice, there is no influence of grooming severity. A BF_{10} greater than one is in favour of a correlation and vice versa. The further the BF_{10} is from one, the greater the evidence. A BF_{10} greater than 3 (or lower than $1/3$) is commonly considered a significant evidence. OCI-R : Obsessive Compulsive Inventory Revised. YBOCS : Yale-Brown Obsessive-Compulsive Scale. BDI : Beck Depression Inventory. STAI : Spielberger's State (A) – Trait (B) Anxiety Inventory. BIS-10 : Barratt impulsivity scale. fNART : French National Adult Reading Test. SSC : Spontaneous Strategy Change. SCAPE : Strategy Change After a Probabilistic Error.”

13. The authors use the term “animal model of compulsive behavior” throughout the manuscript. I suggest to change this to “animal model for compulsive-like behavior” since this animal model is used to study behavioral phenomena modeling human behavior (but not implementing the same disorder).

We followed the reviewer's recommendations and modified the text accordingly lines 33, 70, 72 and 79.

Reviewer #5 (Remarks to the Author):

*Although it makes sense to conceptualize compulsivity as impaired cognitive and behavioral flexibility, reversal learning studies so far showed inconsistent outcomes. Nevertheless, neuronal (OFC) reversal learning impairments are more consistently found in OCD patients and their relatives suggesting inflexibility is a useful neurobiological endophenotype to further investigate mechanistically in animal compulsivity models. The authors present an innovative and well-designed cross-species study using a comparable reversal learning tasks in human OCD patients and *Sapap3* KO mice. Their study is important, first, in implicating the feasibility and validity of a cross-species compulsivity paradigm. In addition, their results show that a subgroup of compulsive (checking OCD) patients and mice are more behaviorally inflexible, however, being more response labile rather than perseverative. This could suggest a more insecure response pattern as an important cross-species mechanism of compulsivity.*

Some minor comments:

- 1. Please address the confounding factor that OCD patients were medicated with serotonergic, dopaminergic or GABA-ergic drugs and mice not.*

As recommended, we discussed this point in the discussion, line 557:

“A potential limitation of our study that can be pointed out is the difference in terms of medication between the two species. Indeed, while our *Sapap3* KO mice were free of any pharmacological treatment that could alter their performance, OCD patients were largely on serotonergic treatment. It is thus logical to think that the results obtained may reflect the effect of the treatment and not their disorder. However, we were able to show the absence of influence of serotonergic medication on performance in our task with no difference between medicated and unmedicated patients. Further, it has been shown that chronic administration of a selective serotonin reuptake inhibitor reduces perseveration and promotes a win-stay strategy in this type of task⁸⁶, which is contrary to our results. We can therefore be confident of the cross-species validity of these findings.”

However, we have also shown with further analyses that patients’ medication does not have an impact on performance in our task both by correlation analysis and by comparison of treated and untreated patients (data added in supplementary results line 54). Therefore, we

believe that the difference of medical treatment between human and mice does not result in significant limitations to the interpretation of our results.

- 2. Although it is great that the animals were tested in a prolonged ecological setting to minimize the influence of random stress and environmental confounds, wouldn't it have been more optimal to adopt something comparable (eg weekly repetitions) for the human testing given that human compulsivity is known to be highly responsive to situational stress?*

We completely agree with this very interesting comment, the answer being necessarily yes since the repetition of the measurement would limit the influence of the stress inherent to the laboratory conditions. However, this same repetition is also a confounding factor in the interpretation of results in humans because of our ability to learn the task and thus improve our performance from one session to the next. Thus, one way to evaluate human performance in a more naturalistic approach would be to evaluate them in their own environment, which they are familiar with and therefore less subject to stress; for example, by using smartphone based apps (Klindt et al., 2017).

- 3. Please explain more how to interpret the excessive response lability in a subtype of the Sapap3 KO mice. The results and author's interpretation suggest these mice behave more like OCD-checkers. However, these mice also display excessive washing/grooming which is more similar to the non-checking OCD-washers.*

We indeed agree with the reviewer that it is counter-intuitive at first glance that a subgroup of Sapap3 KO mice, which could be considered more as an analogous of OCD "washers" subtype, is similar in performance to OCD "checkers" patients. But, considering that it is not uncommon for a patient to present a hybrid compulsive symptomatology mixing both compulsive checking and washing (Fontenelle et al., 2005), it is possible that the impaired subgroup of mice also presents checking behaviours (with response lability being one possible expression of it). We discussed further this point line 540:

"In mice, we cannot conclude that the isolated subgroup of Sapap3 KO mice is analogous to the "checking" subtype of OCD. However, considering that it is not uncommon for a patient to present a hybrid compulsive symptomatology mixing both compulsive checking and washing⁸⁰, one can imagine that the impaired subgroup of mice also presents a mix of compulsive grooming and checking behaviours. Obviously, in mice these checking behaviours are not directly observable, but it would be interesting to test if uncertainty monitoring and checking behaviours are also affected in Sapap3 KO mice. Indeed, one could expect that abnormal increase of uncertainty after reversal would provoke the excessive lability of their behaviour (e.g. with mice over-checking if the previously rewarded stimulus is still valid)."

References

- Bari, A., Theobald, D.E., Caprioli, D., Mar, A.C., Aidoo-Micah, A., Dalley, J.W., and Robbins, T.W. (2010). Serotonin Modulates Sensitivity to Reward and Negative Feedback in a Probabilistic Reversal Learning Task in Rats. *Neuropsychopharmacology* 35, 1290–1301.
- Cohen, J. (1960). A Coefficient of Agreement for Nominal Scales. *Educ. Psychol. Meas.* 20, 37–46.
- Dienes, Z. (2011). Bayesian Versus Orthodox Statistics: Which Side Are You On? *Perspect. Psychol. Sci.* 6, 274–290.
- Fontenelle, L.F., Mendlowicz, M.V., and Versiani, M. (2005). Clinical subtypes of obsessive-compulsive disorder based on the presence of checking and washing compulsions. *Braz. J. Psychiatry* 27, 201–207.
- Gelman, A. (2016). Bayesian inference completely solves the multiple comparisons problem «Statistical Modeling, Causal Inference, and Social Science.
- Gelman, A., and Tuerlinckx, F. (2000). Type S error rates for classical and Bayesian single and multiple comparison procedures. *Comput. Stat.* 15, 373–390.
- Gruner, P., and Pittenger, C. (2017). Cognitive inflexibility in Obsessive-Compulsive Disorder. *Neuroscience* 345, 243–255.
- Harrell Jr, F.E. (2019). Why a Bayesian Approach to Drug Development and Evaluation?
- Hauser, T.U., Iannaccone, R., Dolan, R.J., Ball, J., Hättenschwiler, J., Drechsler, R., Rufer, M., Brandeis, D., Walitza, S., and Brem, S. (2017). Increased fronto-striatal reward prediction errors moderate decision making in obsessive-compulsive disorder. *Psychol. Med.* 47, 1246–1258.
- de Jong, T. (2019). A Bayesian Approach to the Correction for Multiplicity (PsyArXiv).
- Kanen, J.W., Ersche, K.D., Fineberg, N.A., Robbins, T.W., and Cardinal, R.N. (2019). Computational modelling reveals contrasting effects on reinforcement learning and cognitive flexibility in stimulant use disorder and obsessive-compulsive disorder: remediating effects of dopaminergic D2/3 receptor agents. *Psychopharmacology (Berl.)* 236, 2337–2358.
- Klindt, D., Devaine, M., and Daunizeau, J. (2017). Does the way we read others' mind change over the lifespan? Insights from a massive web poll of cognitive skills from childhood to late adulthood. *Cortex* 86, 205–215.
- Landis, J.R., and Koch, G.G. (1977). The Measurement of Observer Agreement for Categorical Data. *Biometrics* 33, 159.
- Lemoine, N.P. (2019). Moving beyond noninformative priors: why and how to choose weakly informative priors in Bayesian analyses. *Oikos* 128, 912–928.

Leopold, R., and Backenstrass, M. (2015). Neuropsychological differences between obsessive-compulsive washers and checkers: A systematic review and meta-analysis. *J. Anxiety Disord.* *30*, 48–58.

Neath, A.A., Flores, J.E., and Cavanaugh, J.E. (2018). Bayesian multiple comparisons and model selection. *Wiley Interdiscip. Rev. Comput. Stat.* *10*.

Patzelt, E.H., Kurth-Nelson, Z., Lim, K.O., and MacDonald, A.W. (2014). Excessive state switching underlies reversal learning deficits in cocaine users. *Drug Alcohol Depend.* *134*, 211–217.

Raftery, A.E. (1995). Bayesian Model Selection in Social Research. *Sociol. Methodol.* *25*, 111.

Vousden, G.H., Paulcan, S., Robbins, T.W., Eagle, D.M., and Milton, A.L. (2020). Checking responses of goal- and sign-trackers are differentially affected by threat in a rodent analog of obsessive–compulsive disorder. *Learn. Mem.* *27*, 190–200.

Westfall, P. (1997). A Bayesian perspective on the Bonferroni adjustment. *Biometrika* *84*, 419–427.

REVIEWERS' COMMENTS:

Reviewer #2 (Remarks to the Author):

The authors have addressed my comments satisfactorily in the revised ms and rebuttal.

Reviewer #3 (Remarks to the Author):

The authors have provided elaborate and appropriate responses to our comments. There is only one remaining concern regarding our 2nd comment. Under point 2), we pointed out that the authors' identification of sub-populations in both mice and humans was based on a cluster analysis in mice, but not in humans. In response, the authors performed a cluster analysis in humans as well, which identified an "impaired" cluster of 7 patients (6 of which are checkers). This additional analysis definitely improves the manuscript. However, the actual number of patients identified in this cluster should be mentioned in the main text explicitly to improve transparency, since it is a relatively low number (albeit mainly consisting of checkers, which is good) compared to the total number of 21 checkers in the study. This is crucial information for the reader and enables the reader to form an opinion on the degree of comparability between analyses without consulting the supplemental material.

Reviewer #4 (Remarks to the Author):

The authors have provided elaborate and appropriate responses to our comments. There is only one remaining concern regarding our 2nd comment. Under point 2), we pointed out that the authors' identification of sub-populations in both mice and humans was based on a cluster analysis in mice, but not in humans. In response, the authors performed a cluster analysis in humans as well, which identified an "impaired" cluster of 7 patients (6 of which are checkers). This additional analysis definitely improves the manuscript. However, the actual number of patients identified in this cluster should be mentioned in the main text explicitly to improve transparency, since it is a relatively low number (albeit mainly consisting of checkers, which is good) compared to the total number of 21 checkers in the study. This is crucial information for the reader and enables the reader to form an opinion on the degree of comparability between analyses without consulting the supplemental material.